# Phosphatidylserine receptors enhance SARS-CoV-2 infection

Dana Bohan[1,☯], Hanora Van Ert[1,☯], Natalie Ruggio[1], Kai J. Rogers[1], Mohammad Badreddine[1], José A. Aguilar Briseño[1], Jonah M. Elliff[1], Roberth Anthony Rojas Chavez[1], Boning Gao[2], Tomasz Stokowy[3], Eleni Christakou[3,4], Petri Kursula[3,5], David Micklem[4], Gro Gausdal[4], Hillel Haim[1], John Minna[2], James B. Lorens[3], Wendy Maury[1]*

1 Department of Microbiology and Immunology, University of Iowa, Iowa City, Iowa, United States of America, 2 Hamon Center for Therapeutic Oncology Research, University of Texas Southwestern Medical Center, Dallas, Texas, United States of America, 3 Department of Biomedicine, University of Bergen, Bergen, Norway, 4 BerGenBio ASA, Bergen, Norway, 5 Biocenter Oulu & Faculty of Biochemistry and Molecular Medicine, University of Oulu, Oulu, Finland

☯ These authors contributed equally to this work.
* wendy-maury@uiowa.edu

**Data Availability Statement:** RNAseq data is available at NCBI GEO under the accession number GSE178942.

**Funding:** This study was primarily supported by National Institutes of Health (NIH/NIAID grant R01

## Abstract

Phosphatidylserine (PS) receptors enhance infection of many enveloped viruses through virion-associated PS binding that is termed apoptotic mimicry. Here we show that this broadly shared uptake mechanism is utilized by SARS-CoV-2 in cells that express low surface levels of ACE2. Expression of members of the TIM (TIM-1 and TIM-4) and TAM (AXL) families of PS receptors enhance SARS-CoV-2 binding to cells, facilitate internalization of fluorescently-labeled virions and increase ACE2-dependent infection of SARS-CoV-2; however, PS receptors alone did not mediate infection. We were unable to detect direct interactions of the PS receptor AXL with purified SARS-CoV-2 spike, contrary to a previous report. Instead, our studies indicate that the PS receptors interact with PS on the surface of SARS-CoV-2 virions. In support of this, we demonstrate that: 1) significant quantities of PS are located on the outer leaflet of SARS-CoV-2 virions, 2) PS liposomes, but not phosphatidylcholine liposomes, reduced entry of VSV/Spike pseudovirions and 3) an established mutant of TIM-1 which does not bind to PS is unable to facilitate entry of SARS-CoV-2. As AXL is an abundant PS receptor on a number of airway lines, we evaluated small molecule inhibitors of AXL signaling such as bemcentinib for their ability to inhibit SARS-CoV-2 infection. Bemcentinib robustly inhibited virus infection of Vero E6 cells as well as multiple human lung cell lines that expressed AXL. This inhibition correlated well with inhibitors that block endosomal acidification and cathepsin activity, consistent with AXL-mediated uptake of SARS-CoV-2 into the endosomal compartment. We extended our observations to the related betacoronavirus mouse hepatitis virus (MHV), showing that inhibition or ablation of AXL reduces MHV infection of murine cells. In total, our findings provide evidence that PS receptors facilitate infection of the pandemic coronavirus SARS-CoV-2 and suggest that inhibition of the PS receptor AXL has therapeutic potential against SARS-CoV-2.

AI134733 (WM), NIH/NCI grants P50 CA070907 (JM), U54 CA260560 (JM)) and a contract from BerGenBio to WJM. DB was supported by National Institutes of Health grant T32AI007511. HVE and JME were supported by National Institutes of Health T32 GM007337. EC was supported by the Norwegian Research Council Industrial PhD Studentship 311399. Study design, data collection and analysis and preparation of the manuscript was coordinated with scientists at BerGenBio. BerGenBio was partially responsible for funding this work. A decision to publish was independent of any input from BerGenBio personnel. The funders had no role in study design, data collection, data analysis, decision to publish, or preparation of the manuscript.

**Competing interests:** I have read the journal's policy and the authors of this manuscript have the following competing interests: GG, DM, and EC are employees of BerGenBio ASA, a company with financial interests in this field. JBL is a former employee of BerGenBio ASA. Partial funding was provided by BerGenBio ASA. JM receives licensing royalties from the NIH and UTSW for distribution of human tumor lines. No other authors have competing interests to declare.

## Author summary

Phosphatidylserine (PS) receptors bind PS and mediate uptake of apoptotic bodies. Many enveloped viruses utilize this PS/PS receptor mechanism to adhere to and internalize into the endosomal compartment of cells. For viruses that have a mechanism(s) of endosomal escape, apoptotic mimicry is a productive route of virus entry. This clever use of this uptake mechanism by enveloped viruses is termed apoptotic mimicry. We evaluated if PS receptors serve as cell surface receptors for SARS-CoV-2 and found that the PS receptors, AXL, TIM-1 and TIM-4, facilitated virus infection when the SARS-CoV-2 cognate receptor, ACE2, was present. Consistent with the established mechanism of PS receptor utilization by other viruses, PS liposomes competed with SARS-CoV-2 for binding and entry. PS is readily detectable on the surface of SARS-CoV-2 virions, and contrary to prior reports we were unable to identify any interaction between AXL and SARS-CoV-2 spike. Pharmacological inhibition of AXL activity and knockout of AXL expression suggest it is the preferred PS receptor during SARS-CoV-2 entry. We propose that AXL is an under-appreciated but potentially important host factor facilitating SARS-CoV-2 entry.

## Introduction

Severe Acute Respiratory Syndrome Coronavirus 2 (SARS-CoV-2) emerged in late 2019 and quickly spread around the world, resulting in the current public health pandemic. SARS-CoV-2 is a beta coronavirus of the sarbecovirus subgenus and is closely related to SARS-CoV, the agent responsible for an epidemic in 2003. SARS-CoV-2 is effectively transmitted between humans and has infected more than 235 million individuals and caused more than 4.8 million deaths worldwide as of October 5, 2021 (WHO). Fortunately, a herculean scientific effort has resulted in the development of SARS-CoV-2 vaccines which have been shown to be efficacious, potentially stemming the pandemic. Nonetheless, in combination with vaccines, continued development of efficacious antivirals is needed, as outbreaks continue in under-vaccinated regions and severe disease caused by SARS-CoV-2 variants of concern is not eradicated following vaccination. Towards this goal, a more comprehensive understanding of SARS-CoV-2 interactions with host cells will be required.

SARS-CoV-2 entry into cells is mediated by the viral spike glycoprotein (S) binding to Angiotensin Converting Enzyme 2 (ACE2) [1–3]. The S1 subunit of S binds to ACE2 while S2 mediates membrane fusion. Cleavage at the S1/S2 junction occurs during virus egress from producer cells by the host protease furin which facilitates S1 binding to ACE2. A second site termed S2' is also cleaved by the host proteases. Cleavage by TMPRSS2 at the cell surface promotes fusion of the viral and host plasma membranes [2,4]. Alternatively, SARS-CoV-2 virions can be internalized via clathrin-mediated endocytosis after ACE2 binding, wherein host cathepsins (especially cathepsin L) proteolytically cleave S2 to S2' [3,5–7]. Adherence factors that enhance virion binding and increase infectivity have also been identified, namely neuropilin 1 and heparan sulfate [8,9].

Binding and internalization of a variety of different enveloped viruses occurs through virion associated phosphatidylserine (PS) binding to host PS receptors. Members of the TIM family (TIM-1 and TIM-4) bind PS directly while another family of PS receptors, the TAM tyrosine kinase receptor family (TYRO3, AXL and MERTK), bind PS indirectly through the adaptor proteins Gas6 and Protein S. These PS receptor mediate binding and internalization of a wide range of viruses, including filoviruses, alphaviruses, and flaviviruses [10–13]. TIM-1, TIM-4,

and AXL appear to be the most efficacious at mediating viral entry given their prevalent use among enveloped viruses [14–16]. Once virions are within the endosome, events that result in virion fusion with cellular membranes are virus specific, with filoviruses requiring viral glycoprotein processing followed by interactions with Niemann Pick C1 protein (NPC1) to initiate fusion, whereas flaviviruses rely on endosomal acidification driving glycoprotein conformational changes which mediate fusion [17,18].

Given that PS receptors mediate entry of other enveloped viruses through interactions with viral membrane PS, we assessed the role of PS receptors on SARS-CoV-2 infection and the mechanism of interaction. We found that plasma membrane-expressed PS receptors by themselves do not result in productive SARS-CoV-2 infection; however, these receptors enhance infection when low levels of ACE2 are expressed. Our findings indicate that these receptors potentiate ACE2-dependent SARS-CoV-2 entry through PS-dependent interactions. Appreciation of this route of entry provides an additional pathway that could be therapeutically targeted to inhibit virus entry and subsequent infection.

## Results

### PS receptors enhance ACE2-dependent SARS-CoV-2 infection

The ability of TIM and TAM family PS receptors to support SARS-CoV-2 infection were initially examined in transfected HEK 293T cells. Wild-type HEK 293T cells do not express significant amounts of ACE2 or PS receptors and are poorly susceptible to SARS-CoV-2 infection [3,19]. Expression plasmids encoding ACE2 and/or the PS receptors, AXL or TIM-1, were transfected, resulting in expression of these receptors on the surface of the transfected cells (**S1A and S1B Fig**). Dual transfection did not alter expression of ACE2 or PS receptors relative to single transfection. Low levels of transfected ACE2 plasmid (50–250 ng) supported modest increases in infection of SARS-CoV-2 and vesicular stomatitis virus pseudovirions bearing SARS-CoV-2 spike (VSV/Spike) (**Figs 1A and S1C**). In contrast, PS receptors AXL and TIM-1 by themselves did not facilitate infection of SARS-CoV-2. However, when low levels of ACE2 were co-expressed with either AXL or TIM-1, the combinations enhanced infection over that observed with ACE2 alone. These data suggested that these PS receptors can potentiate ACE2-dependent SARS-CoV-2 infection.

To further assess the ability of the PS receptors to facilitate ACE2-dependent infection, the amount of transfected PS receptor plasmid was held constant with increasing concentrations of ACE2. While both PS receptors enhanced ACE2-dependent virus infection, TIM-1 enhanced recombinant replication-competent VSV that encoded SARS-CoV-2 spike (rVSV/Spike) [20] infection over a wider range of ACE2 concentrations than AXL did, with AXL consistently enhancing infection only at 250 ng of transfected ACE2 plasmid (**Figs 1B, 1C and S1C**). The more limited ability of AXL to potentiate was not due to limiting Gas6 in the media as the addition of Gas6 to media did not enhance the effect. At higher concentrations of ACE2 plasmid, the enhancement of infectivity mediated by PS receptors was reduced, with no PS receptor enhancement observed when 1 μg of ACE2 plasmid was transfected. Thus, only when ACE2 is limiting on the cell surface do PS receptors facilitate infection.

Consistent with a role for PS receptors in ACE2-dependent SARS-CoV-2 entry, we observed greater virion attachment to cells when PS receptors were exogenously expressed in HEK 293T cells (**Fig 1D**). PS receptor expression also increased internalization of FITC-labeled virions relative to vector-transfected controls (**Fig 1E**). As expected, ACE2 was also able to internalize virions, but unexpectedly to a lesser extent than the PS receptors.

PS receptors, TIM-4, TYRO3, and MerTK, were also examined for their ability to increase pseudovirus infection. TIM-4 enhanced ACE2-dependent entry of VSV/Spike in a manner

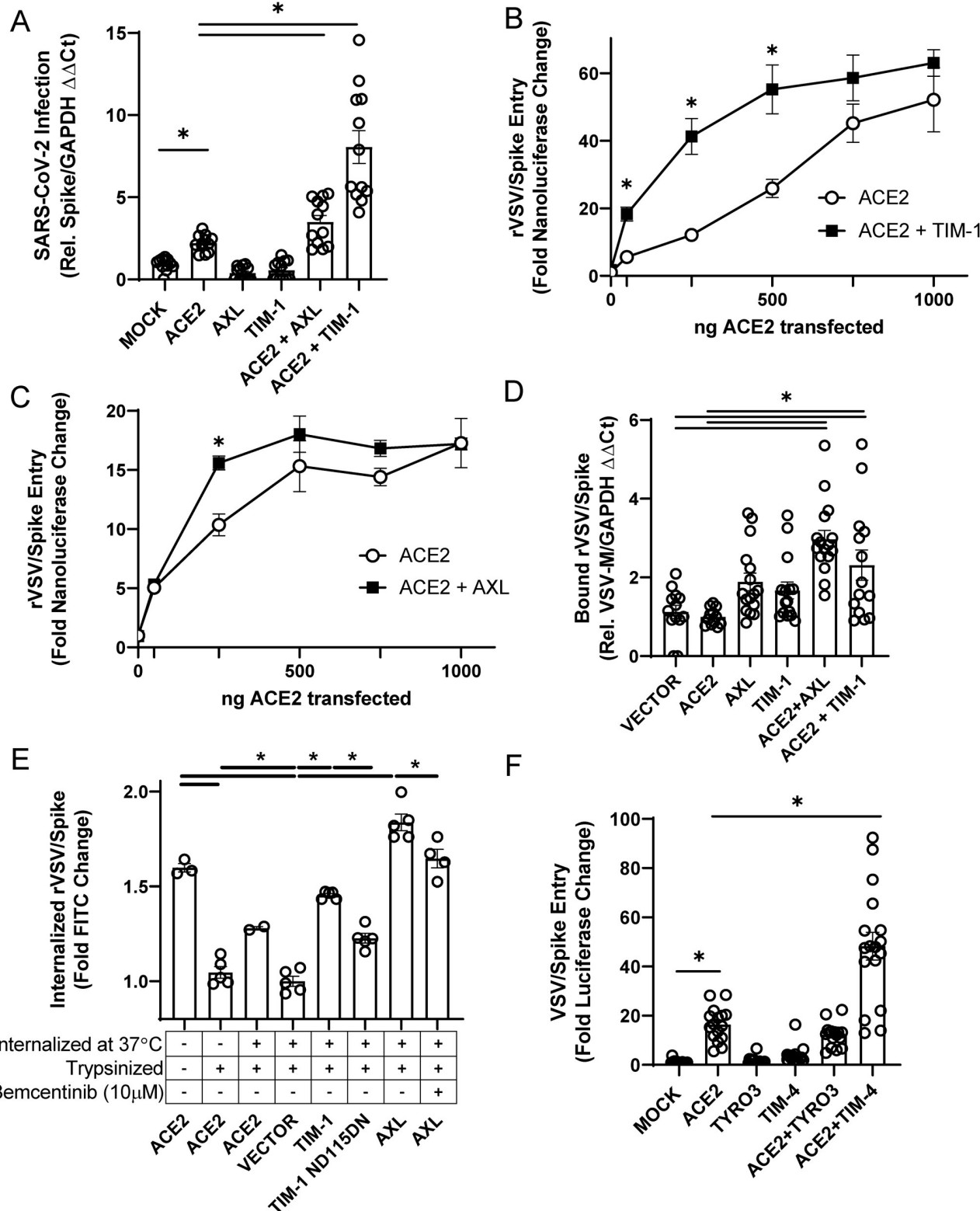

**Fig 1. PS receptors synergize with ACE2, enhancing SARS-CoV-2 infection of HEK 293T cells. A)** Cells transfected with PS receptor expression plasmids, AXL or TIM-1, with or without 50 ng of ACE2 and infected 48 hours later with SARS-CoV-2 (MOI = 0.5). Resulting cellular infection was determined by viral loads 24 hours after initial infection using RT-qPCR. **B-C)** PS receptors, TIM-1 (**B**) and AXL (**C**), enhance rVSV/Spike infection at low concentrations of transfected ACE2. **D)** Virus binding of cells transfected with PS receptor plasmids with or without 50 ng of ACE2. rVSV/Spike

was bound to transfected cells at 48 h following transfection and bound virus was measured via RT-qPCR. **E)** PS receptors mediate internalization of rVSV/Spike. Virion internalization was measured at 24 h after transfection with 1 μg of the indicated plasmids. FITC-labeled rVSV/Spike was bound for 1-hour, unbound virions washed away, and cells shifted to 37˚C for 30 minutes. Non-internalized virus was then cleaved from cell surface by trypsin. Cells were washed, and FITC retention quantified by flow cytometry. **F)** HEK 293T cells transfected with PS receptor plasmids, TYRO3 or TIM-4, with or without 250 ng of ACE2 and infected 48 hours later with VSV/Spike. Viral loads were determined 24 hours following infection. Data shown are pooled from at least 3 independent experiments (**A**, **B**, **C**, **D**, **E**, **F**). Data represented as means ± SEM. Student's t-test (**A, E**) and multiple t-test (**B, C**), One-Way ANOVA with multiple comparisons (**D, F**); asterisks represent p < 0.05.

similar to TIM-1; however, TYRO3 and MerTK of the TAM family did not mediate increased entry, despite expression on the plasma membrane expression after transfection (**Figs 1F, S1A, S1B and S1D**).

The enhancement of ACE2-dependent infection by PS receptors was specific for SARS-CoV-2 as infection with VSV-luciferase pseudovirions bearing Lassa virus GP was not affected by expression of these receptors (**S1E Fig**). These studies indicate that PS receptors enhance the binding and internalization of spike-bearing virions and potentiate SARS-CoV-2 infection when ACE2 is present. Further, these data provide evidence that VSV/Spike pseudovirions and rVSV/Spike serve as a useful BSL2 surrogate for SARS-CoV-2 entry events as others have shown [21–23].

## PS receptors bind to virion-associated PS, not the SARS-CoV-2 spike protein

We took several different approaches to examine the mechanism by which PS receptors interact with SARS-CoV-2. In the context of other viral pathogens, PS receptors are known to bind to PS within the outer leaflet of virion membranes and mediate endosomal internalization of virions, shuttling virus to cognate endosomal receptors. Recently, antibodies that bind to PS have been developed [24] and we leveraged this tool to detect PS availability on SARS-CoV-2 virion membranes. UV-inactivated SARS-CoV-2, rVSV/Spike, or adenovirus (Ad5, non-enveloped negative control) were coated on ELISA plates and, using the anti-PS antibody bavituximab, PS was readily detectable on both SARS-CoV-2 and rVSV/Spike virions, but not on Ad5 virions (**Fig 2A**). Interestingly, at higher concentrations of virions on ELISA plates, significantly greater levels of PS were detected on SARS-CoV-2 virions compared to rVSV/Spike virions. To demonstrate the specificity of the PS ELISA, PS liposomes that compete with virions for binding to PS receptors [25] effectively competed for bavituximab, thereby reducing antibody binding to the plate-bound virus (**S2A Fig**). To determine the effect of liposomes on virus infection, increasing concentrations of PS or PC liposomes were evaluated for their ability to compete with virus for PS binding sites in ACE2 + TIM-1 or ACE2 + AXL transfected HEK 293T cells. PS liposomes effectively blocked rVSV/Spike infection, whereas PC liposomes were significantly less effective (**Fig 2B and 2C**). We also assessed the activity of a TIM-1 mutant, ND115DN which has a disrupted TIM-1 PS binding pocket, for its ability to facilitate rVSV/Spike entry. ND115DN was expressed at equivalent levels as WT TIM-1 following HEK 293T cell transfection (**S2B Fig**). This TIM-1 mutant exhibited reduced internalization of FITC-labeled rVSV/Spike relative to WT TIM-1 (**Fig 1E**) and did not enhance ACE2-dependent infection (**Fig 2D**), indicating that the TIM-1 PS binding pocket is critical for these activities [12,19].

Others have reported that the N-terminal domain of SARS-CoV-2 spike directly binds to AXL and is important for AXL-mediated entry of SARS-CoV-2 [26]. To assess spike/AXL interactions, purified SARS-CoV-2 spike ectodomain-Fc, spike receptor binding domain-Fc (RBD) or spike N-terminal domain-Fc (NTD) was incubated with HEK 293T cells transiently expressing AXL. Flow cytometry was used to detect spike proteins bound to AXL. Parallel

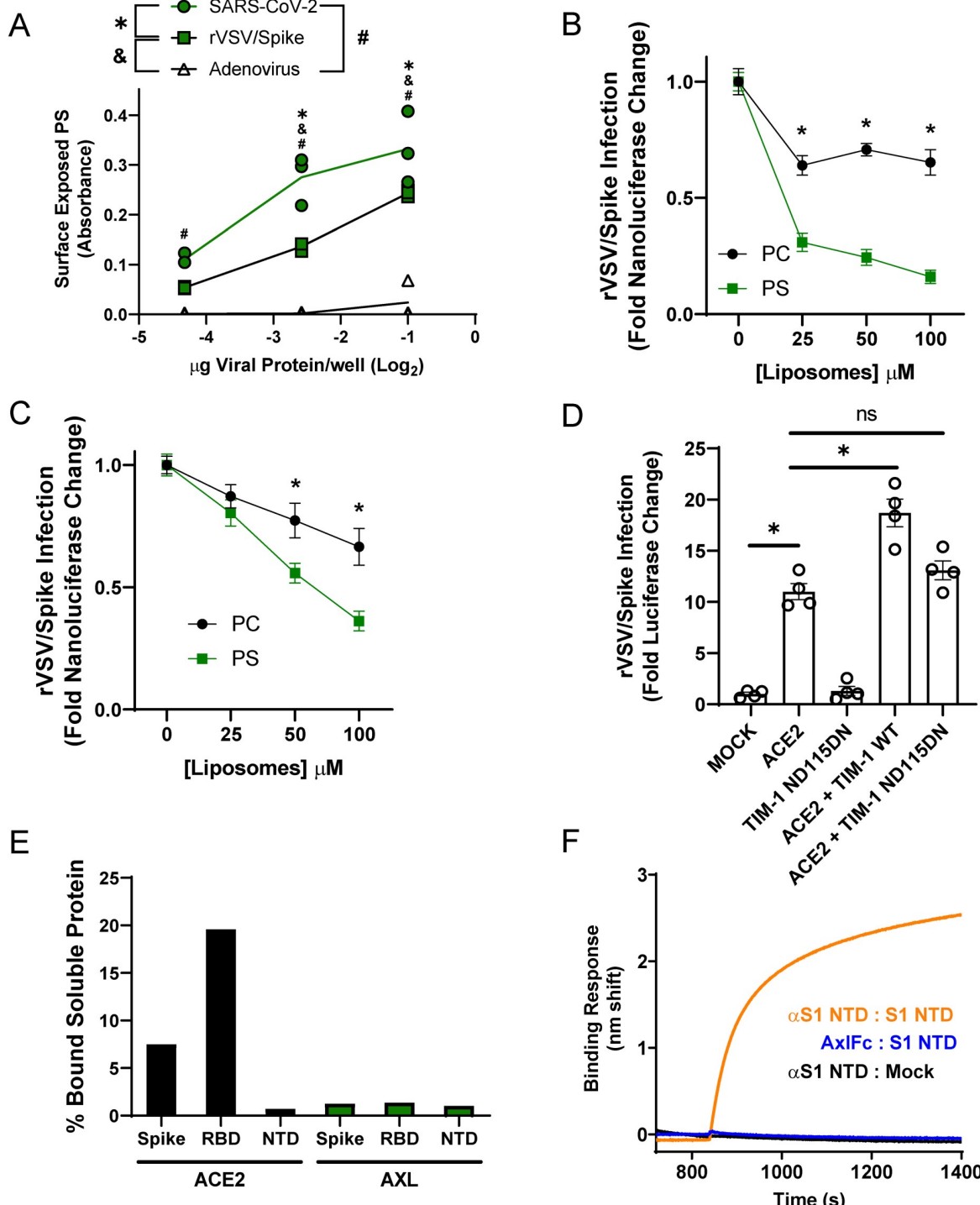

**Fig 2. PS receptors interact with SARS-CoV-2 by binding to virion PS. A)** PS is readily detectable on UV irradiated SARS-CoV-2 virions and rVSV/Spike. Indicated quantities of viral particles (determined by protein content) were coated in ELISA plates, and PS was detected using bavituximab followed by secondary antisera. **B-C)** PS liposomes interfere with rVSV/Spike infection. HEK 293T cells transfected with 50 ng of ACE2 plasmid and 1 μg of TIM-1 (**B**) or AXL (**C**) plasmid. Cells were infected with rVSV/Spike in the presence of increasing concentrations of PS or PC liposomes and assessed for nanoluciferase activity 24 hours later. **D)** HEK 293T cells were transfected with 1 μg of plasmid expressing WT or PS binding pocket mutant TIM-1 (ND115DN) with or without 250 ng of ACE2 plasmid and infected 48 hours later with rVSV/Spike. Luminescence fold change were compared to Mock transfected lysates that were set to a value of 1. **E)** Surface expressed AXL is unable to directly interact with purified SARS-CoV-2 spike/Fc proteins. HEK 293T cells transfected with AXL or ACE2 were incubated with soluble Spike protein-Fc, S1 RBD-Fc or S1 NTD-Fc and subsequently incubated with

an Alexa 647 secondary against Fc. Transfected cells bound to spike constructs were detected by flow cytometry. **F)** Purified AXL does not bind to the NTD of SARS-CoV-2 spike. Biolayer interferometry association curves show that immobilized AXL-Fc fails to interact with purified NTD of spike. Data are pooled from at least 3 independent experiments (**B**, **C**) or are representative of at least 3 experiments (**A**, **D**, **E**, **F**). Data represented as means (or individual datapoints) ± SEM. Multiple t-test (**B**, **C**), One-way ANOVA with multiple comparisons (**A**, **D**); asterisks represent p < 0.05.

ACE2 binding to soluble SARS-CoV-2 spike served as a positive control. As the spike NTD-Fc was not expected to bind to ACE2, an ELISA confirmed the ability of a conformationally dependent α-spike NTD monoclonal antibody to bind NTD-Fc, suggesting that NTD-Fc was in its native conformation (**S2D Fig**). The full-length spike-Fc and the RBD-Fc bound to ACE2, but no interactions were detected between any of the purified spike proteins and AXL (**Fig 2E**) despite evidence of robust AXL surface expression on transfected HEK 293T cells (**S2C Fig**) and the equivalent levels of detection of the purified proteins via ELISAs (**S2E Fig**). Biolayer interferometry studies confirmed and extended our findings that recombinant AXL does not bind to purified NTD, whereas NTD interaction with the α-spike NTD monoclonal antibody was readily detected (**Fig 2F**). Thus, using two complementary approaches, we were unable to demonstrate direct interactions of AXL with spike. In total, our studies are consistent with PS receptors interacting with SARS-CoV-2 virions through the well-established mechanism of virion-associated PS binding to TIM-1 and AXL.

## Redundant routes of virus entry: endosomal vs. plasma membrane mediated infection

ACE2-dependent coronaviruses enter cells through two different routes: 1) An endosomal route of virus uptake that requires low pH-dependent cysteine protease, cathepsin L, processing of spike and 2) a plasma membrane route that is dependent upon serine protease, TMPRSS2, cleavage of spike [3,27]. Others have reported that TMPRSS2-dependent entry is preferentially utilized by the virus when this protease is expressed [28]. We examined the route of virus entry at play when ACE2, PS receptors and/or TMPRSS2 was expressed.

Initially, we assessed enhancement conferred by increasing amounts of TMPRSS2 plasmid on ACE2-dependent infection. As anticipated, we found that low levels (10 ng) of TMPRSS2 expressing plasmid enhanced VSV/Spike pseudovirion entry in HEK 293T cells co-transfected with 50 ng of ACE2 expressing plasmid (**Fig 3A**). However, at higher concentrations of TMPRSS2 plasmid, TMPRSS2 did not enhance infection, perhaps due to excessive protease activity. We also evaluated the requirement for TMPRSS2 expression with increasing concentrations of ACE2. Transfection of 10 ng of TMPRSS2 plasmid enhanced virus infection at low concentrations of ACE2, but virus entry became TMPRSS2-independent at higher levels of ACE2 in a manner similar to the effects of the PS receptors. Taken with Fig 1B and 1C, these studies indicate that the PS receptors and TMPRSS2 can facilitate ACE2-dependent virus infection when ACE2 is limiting, but with increasing ACE2 concentrations the infections become independent of these entry factors. This may be related to effects of soluble ACE2 (sACE2) on entry, as recent reports show that excessive ACE2 expression facilitates sACE2 cleavage. This soluble protein can bind to both SARS-CoV-2 and receptors of the renin-angiotensin system [29]. Alternatively, high ACE2 expression may obviate any rate or accessibility advantages that TMPRSS2 or PS receptors offer.

To evaluate how PS receptors and/or TMPRSS2 expression alter the route of ACE2-dependent infection, HEK 293T cells were transfected PS receptors and incubated with VSV/Spike in the presence or absence of the cysteine protease inhibitor, E-64, that blocks endosomal cathepsin activity. Non-toxic levels of E-64 blocked ACE2-dependent entry (**Figs 3C and S3**),

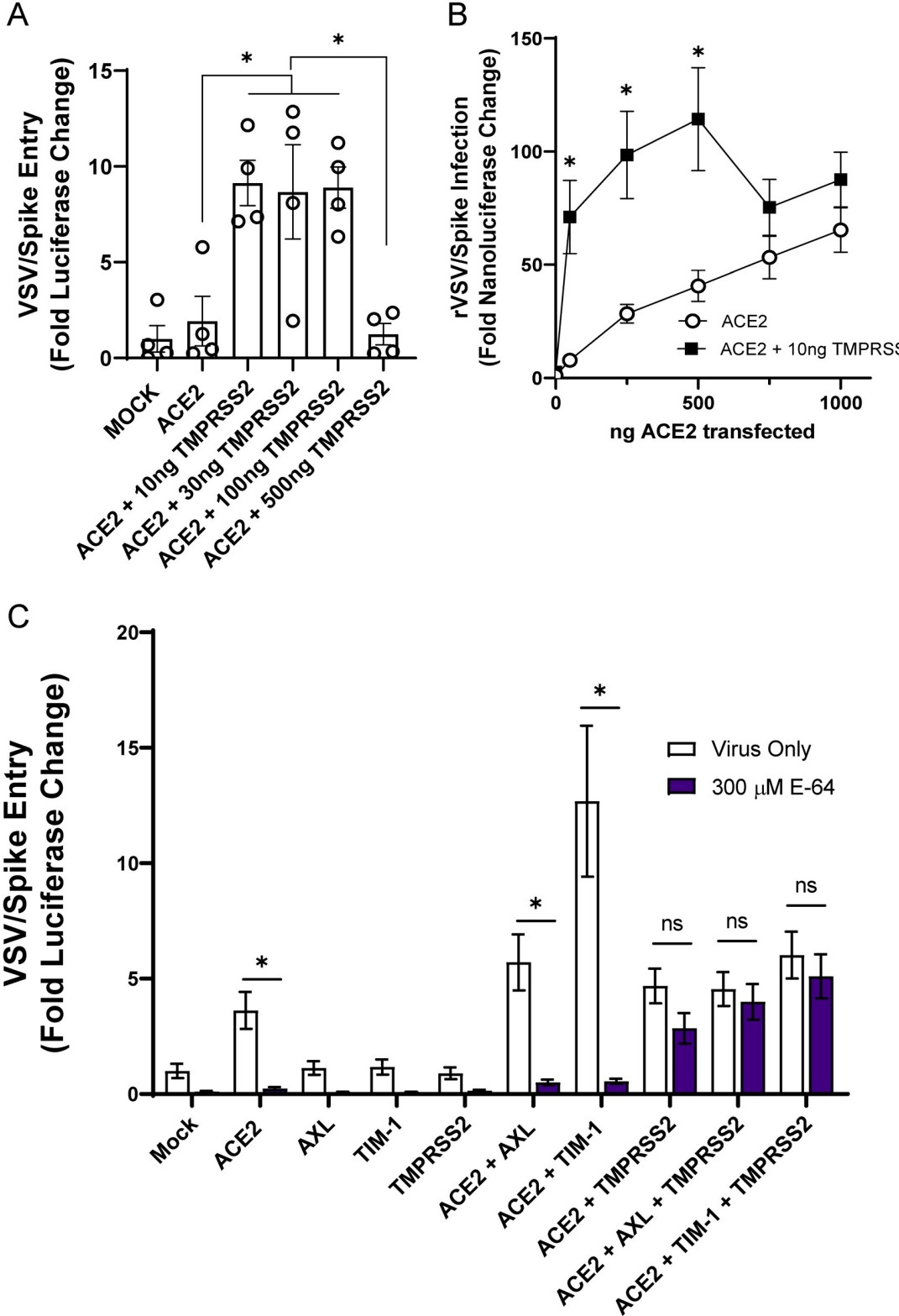

**Fig 3. The route of SARS-CoV-2 entry is altered by TMPRSS2 expression. A)** HEK 293T cells were transfected with 50 ng ACE2 plasmid and TMPRSS2 plasmid as noted and infected at 48 h with VSV/Spike. At 24 hpi, luminescence activity was determined. Findings are shown relative to empty vector (Mock) transfected cells. **B)** TMPRSS2 expression enhances rVSV/Spike infection at low levels of ACE2 expression. HEK 293T cells were transfected as indicated and rVSV/Spike infection

assessed by measuring luminescence activity at 24 hpi. **C)** Transfected HEK 293T cells were transfected and infected with VSV/Spike at 48 h in the presence or absence of E-64 (300 μM). Luciferase activity was determined 24 hpi. Data are pooled from at least 3 independent experiments (**B**, **C**) or are representative of at least 3 experiments (**A**). Data represented as means ± SEM. Student's t-tests (**A**) Multiple t-tests (**B**), Two-way ANOVA with row-wise multiple comparisons (**C**); asterisks represent $p < 0.05$.

indicating that virions were entering these cells in a cysteine protease-dependent manner, likely through the endosomal compartment. The enhancement of virus entry conferred by the combination of PS receptors and ACE2 was dramatically inhibited by E-64, providing evidence that this is the route of virion uptake that is enhanced by PS receptors. These findings are consistent with earlier reports that PS receptors mediate enveloped virus and apoptotic body internalization into the endosomal compartment [12,13,30]. In cells that expressed ACE2 and TMPRSS2, virus entry was no longer sensitive to E-64 as previously reported [6,28]. VSV/Spike entry in the presence of TMPRSS2, PS receptors, and ACE2, was also insensitive to E-64, suggesting that the TMPRSS2 activity mediates entry at the plasma membrane. Further, the presence of PS receptors did not enhance virus infection in the presence of TMPRSS2, suggesting that this route of virus entry is independent of PS receptor utilization.

## Inhibition of endogenous AXL utilization blocks SARS-CoV-2 entry

We next evaluated the ability of endogenously expressed PS receptors to enhance SARS-CoV-2 in ACE2-positive cells. Vero E6 cells that express ACE2, AXL, and TIM-1 (**S4A and S4B Fig**) were initially assessed [19]. Competition studies using PS liposomes confirmed that PS receptors are important for SARS-CoV-2 infection of these cells, with increasing doses of PS, but not PC, liposomes inhibiting VSV/Spike entry, similar to our findings in transfected HEK 293T cells (**Fig 4A**). PS liposomes also significantly reduced SARS-CoV-2 binding to the surface of Vero E6 cells, implicating endogenous PS receptors in virus attachment (**Fig 4B**). These findings reinforce the importance of either AXL, TIM-1, or both for SARS-CoV-2 entry.

To assess if AXL was important for infection of Vero E6 cells, the selective AXL kinase inhibitor, bemcentinib, was tested for its ability to block SARS-CoV-2 infection. This inhibitor inhibits AXL kinase activity after cargo binding and is thought to reduce internalization of cargo-bound receptors. Concurring with this mechanism, bemcentinib inhibited internalization of FITC-labeled rVSV/Spike in HEK 293T cells (**Fig 1E**). In a dose dependent manner, bemcentinib profoundly inhibited SARS-CoV-2 virus load and VSV/Spike pseudovirion infection at 24 hpi (**Figs 4C and S4D**). Bemcentinib toxicity was tested on human lung epithelial cells and was nontoxic at the concentrations used (**S4C Fig**). In RNAseq studies, SARS-CoV-2 infection (MOI = 0.01) of Vero E6 cells resulted in viral transcripts composing ~80% of all mRNAs at 18 hpi (**Fig 4D**). Consistent with an important role for AXL in SARS-CoV-2 infection, when these cells were treated with 1 μM bemcentinib at the time of infection, the fraction of viral transcripts dropped precipitously, decreasing to ~10% of the total reads. Additional studies with a broad-spectrum TAM inhibitor, BMS-777607, also modestly reduced virus infection in a dose dependent manner (**Fig 4E**), reinforcing our observation that inhibition of TAM family members blocks SARS-CoV-2 infection.

To determine if TIM-1 contributed to SARS-CoV-2 infection of Vero E6 cells, the blocking anti-human TIM-1 monoclonal antibody, ARD5, was evaluated for inhibition of recombinant VSV (rVSV) bearing either Ebola glycoprotein (EBOV GP) or spike. While rVSV/EBOV GP was inhibited by ARD5 as previously reported [12,31], ARD5 had no effect on rVSV/Spike infection (**S4E Fig**). Thus, despite robust expression of both PS receptors, AXL was preferentially utilized for SARS-CoV-2 infection in these cells. While preferential PS receptor

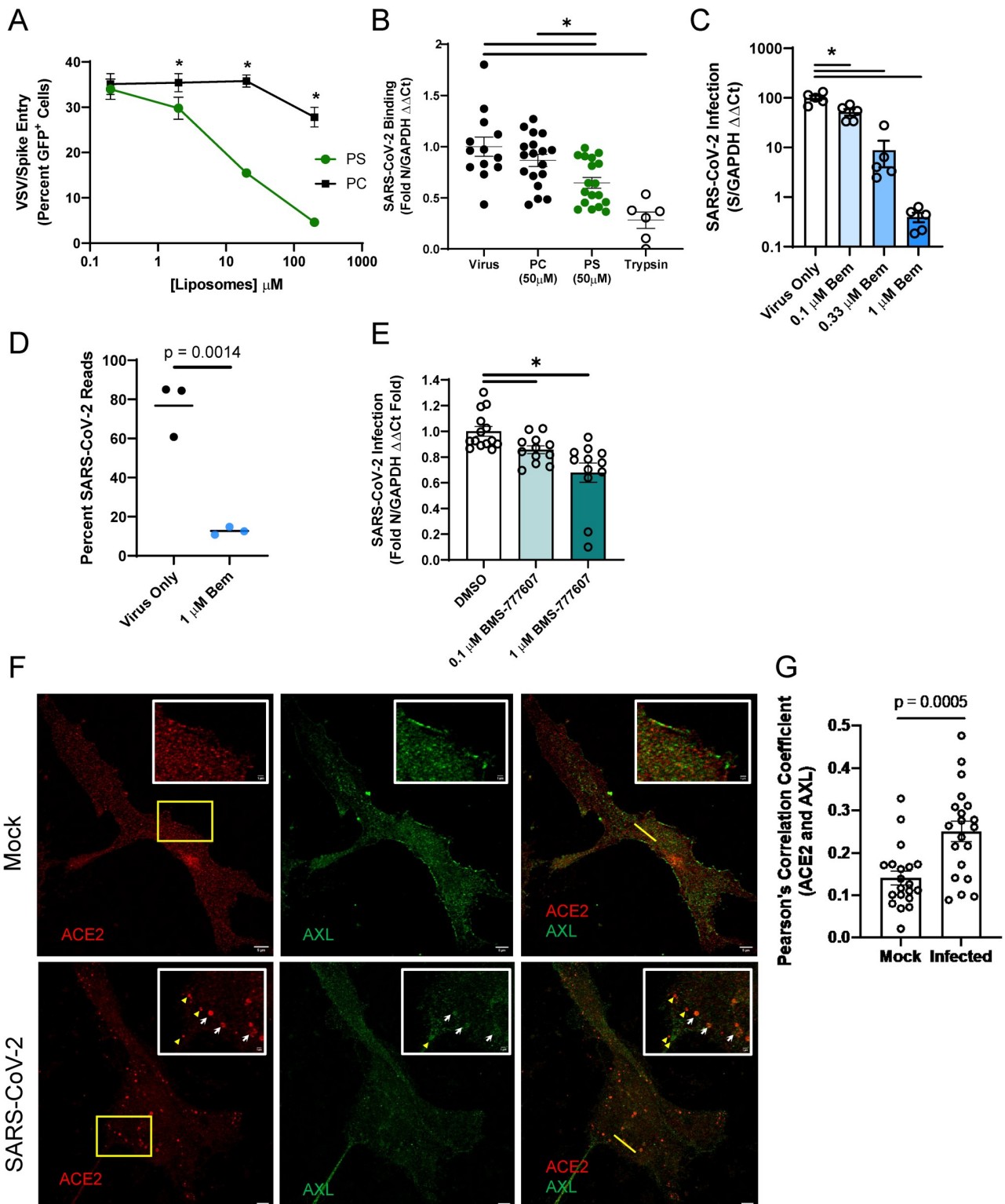

**Fig 4. AXL has a prominent role in SARS-CoV-2 entry into Vero E6 cells.** A) PS liposomes interfere with SARS-CoV-2 pseudovirion entry. Vero E6 cells were treated with increasing concentrations of PS or PC liposomes and infected with VSV/Spike pseudovirions for 24 hours. Infection was detected by GFP fluorescence expressed from the VSV genome. **B)** PS liposomes disrupt SARS-CoV-2 binding. Vero E6 cells were incubated with SARS-CoV-2 (MOI = 5) at 10°C for 1 hour in the presence of indicated liposomes, washed extensively, and viral load assessed by RT-qPCR. **C)** AXL signaling inhibitor bemcentinib inhibits SARS-CoV-2 infection in Vero E6 cells. Cells were treated with bemcentinib and infected with SARS-CoV-2

(MOI = 0.01). Viral loads were measured 24 hpi by RT-qPCR. **E)** RNAseq studies in Vero E6 cells demonstrate bemcentinib inhibition. Cells were treated with 1 μM bemcentinib, infected with SARS-CoV-2 (MOI = 0.01) and mRNA harvested 18 hpi. mRNA was deep sequenced on an Illumina platform, and 'Percent Viral Reads' were calculated by alignment to the SARS-CoV-2 genome. **E)** Broad spectrum TAM inhibitor BMS-777607 inhibits SARS-CoV-2 infection in Vero E6 cells. Cells were treated with inhibitor at indicated concentrations prior to challenged (MOI = 0.01), and viral loads measured 24 hpi by RT-qPCR. **F-G)** Enhanced colocalization of AXL and ACE2 during SARS-CoV-2 infection. STED micrographs shows staining for ACE2 (red) and AXL (green) and merged in Vero E6 cells (**F**). Insets are enlarged images from regions highlighted by yellow rectangles. White arrows indicate shared vesicular structures between the two channels. Yellow arrowheads indicate objects that are only seen in one channel. Plot profiles are shown in **S4F**, representing signal intensity along the yellow lines in the merged panels. Pearson's correlation coefficients of ACE2 and AXL were calculated for n = 20 mock and infected cells (ROI determined by cell borders) (**G**). Data are pooled from at least 3 independent experiments (**B**, **E**) or are representative of at least 3 experiments (**A**, **C**, **F**, **G**). Data are represented as means ± SEM. Multiple t-tests (**A**) Student's t-test (**B**, **C**, **D**, **G**); asterisks represent $p < 0.05$.

utilization has been reported for other pathogens [19], our previous studies indicated that TIM-1 rather than AXL was the preferred receptor, in contrast to our current observations with SARS-CoV-2. Host factors or virion attributes determining PS receptor preference are currently unexplored.

As the bulk of ACE2 in Vero E6 cells is intracellular (**S4B Fig**), surface expressed-AXL may be facilitating SARS-CoV-2 uptake into the endosomal compartment where proteolytic processing and ACE2 interactions mediate fusion of the viral envelope and cellular membranes. Previous studies with the betacoronavirus responsible for the 2003–2004 outbreak, SARS-CoV, demonstrated that spike protein led to ACE2 translocation from the plasma membrane to cytoplasmic compartments, specifically co-localizing with the early endosomal marker EEA1 [32]. Further, at 3 hpi, SARS-CoV antigens colocalized with vesicular ACE2, and ACE2 formed notable vesicular puncta in the infected cells [32]. We utilized Stimulated Emission Depletion (STED) microcopy, leveraging the super resolution capabilities of this platform to investigate ACE2 and AXL colocalization in uninfected and infected Vero E6 cells. In uninfected cells, AXL and ACE2 were found on the plasma membrane and intracellularly, but colocalize poorly (**Figs 4F, 4G** and **S4F**). However, as shown in the micrographs (white arrows) and the associated fluorescence intensity plot profiles (**S4F Fig**, yellow lines in **Fig 4F** highlight selected ROI), ACE2 and AXL demonstrate overlapping localization patterns within large cytoplasmic punctate structures in Vero E6 cell that were infected with SARS-CoV-2 for 24 hours. Pearson's correlation coefficients of infected and uninfected cells calculated for AXL and ACE2 intensity demonstrated a significant increase in colocalization values between this PS receptor and ACE2 in infected cells, relative to mock counterparts. (**Fig 4G**). These data support the possibility that PS receptors, predominantly AXL, enhance SARS-CoV-2 trafficking into these intracellular puncta where ACE2 is abundant. It then follows, and is supported by our data, that bemcentinib-mediated AXL inhibition prevents this internalization.

## AXL promotes SARS-CoV-2 infection in a range of lung cell lines

In addition to ACE2, many human lung cell lines express AXL [26,33] (**S5A Fig**). We evaluated these lines for their ability to support SARS-CoV-2 infection and whether infection was sensitive to bemcentinib inhibition. The panel of lung cells that were selected included A549 (adenocarcinoma) stably expressing ACE2, H1650 (NSCLC), HCC1944 (squamous), HCC2302 (adenocarcinoma) and Calu-3 (adenocarcinoma).

These cells were inoculated with SARS-CoV-2 (MOI = 0.5) in the presence or absence of a serial dilution of the AXL inhibitor, bemcentinib, or the cysteine protease inhibitor, E-64. The cell lines A549$^{ACE2}$, H1650, HCC1944, and HCC2302 readily supported SARS-CoV-2 infection, and viral loads 24 hpi were decreased in a dose-dependent manner, by bemcentinib or E64 (**Fig 5A, 5B, 5C and 5D**). While bemcentinib was efficacious in decreasing viral infection into lung cells, these findings are more modest than the reduction found in Vero E6 cells.

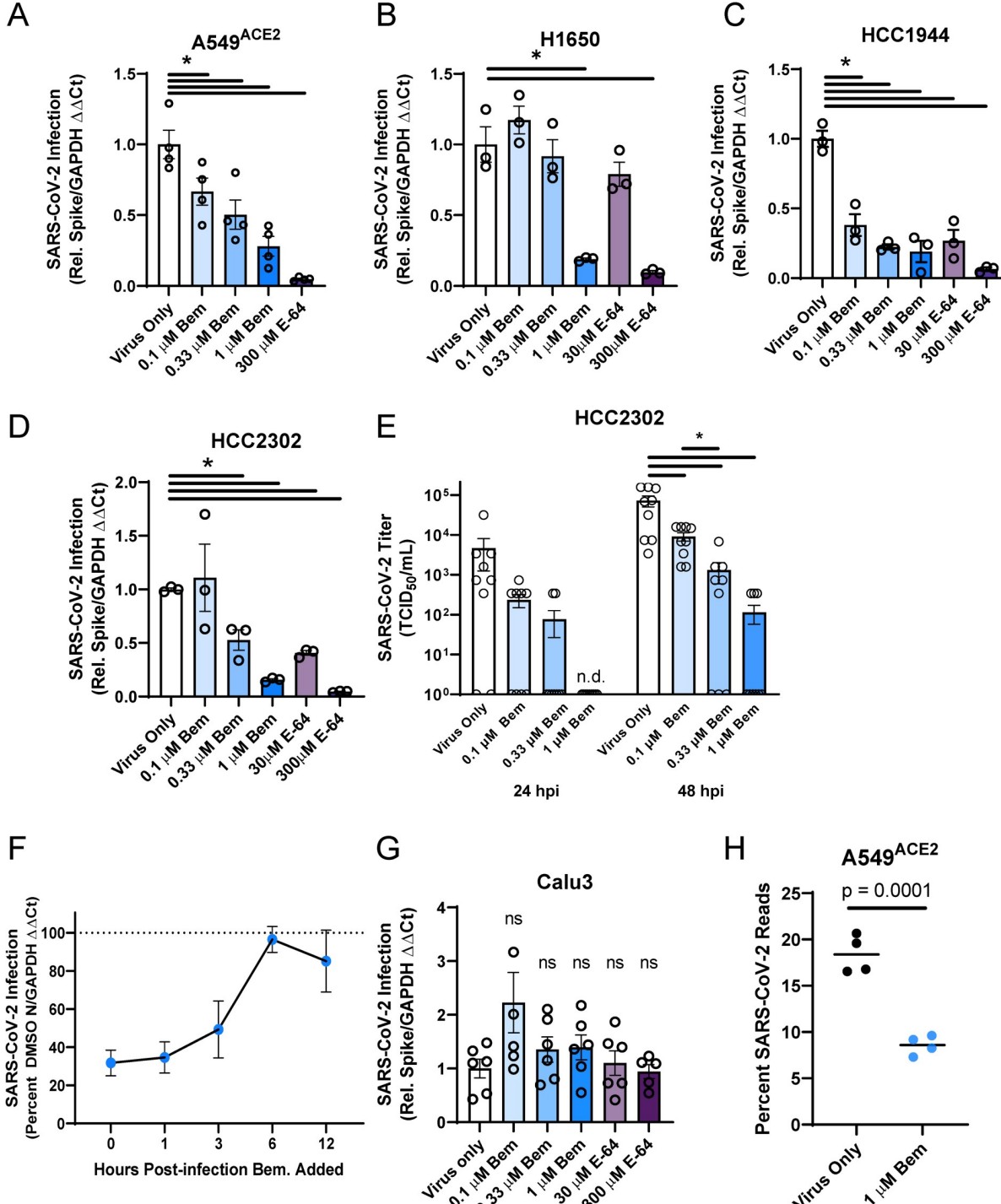

**Fig 5. AXL inhibition reduces SARS-CoV-2 infection in human lung cells. A-E)** SARS-CoV-2 infection (MOI = 0.5) is reduced by AXL inhibition by bemcentinib or E64 in multiple human lung cell lines, including A549$^{ACE2}$(**A**), H1650 (**B**), HCC1944 (**C**), HCC2302 (**D**). Inhibitors were added to cells 1 h prior to infection and maintained on the cells for the entire infection. At 24 hpi, viral load was determined. **E)** Infectious virus produced by HCC2302 cells was inhibited by bemcentinib. HCC2302 cells were treated with bemcentinib at the indicated concentrations and infected with SARS-CoV-2 (MOI = 0.5). Input virus was removed 6 hpi and media containing the appropriate bemcentinib concentration was added. Supernatant was collected at 24 and 48 hpi and titered by TCID$_{50}$ assays on Vero E6-TMPRSS2 cells. TCID$_{50}$/mL was calculated by the Spearmann-Karber method. **F)** Timing of effect bemcentinib inhibition of SARS-CoV-2 infection. In time-of-addition studies, SARS-CoV-2 (MOI = 0.01) was added to infected H1650 cells (human lung cells) to initiate the experiment and 1 μM bemcentinib was added at the times noted. Cells were harvested at 24 hpi for viral load determinations. **G)** Calu-3 are insensitive to

bemcentinib and E64. Studies were performed as described for panels A-E. **H)** A549$^{ACE2}$ were treated with bemcentinib, infected with SARS-CoV-2 (MOI = 0.5) and mRNA harvested 24 hpi. mRNA was deep sequenced and viral loads calculated by alignment to the SARS-CoV-2 genome. Data are representative of at least 3 experiments (**A, B, C, D, E, F, G**). Data represented as means ± SEM. Student's t-test; asterisks represent $p < 0.05$.

Infectious SARS-CoV-2 present in HCC2302 cell supernatants at 24 and 48 hpi were also markedly decreased by bemcentinib (**Fig 5E**), demonstrating that bemcentinib treatment reduced production of new infectious virus in a dose-dependent manner. Further, at 1 μM of bemcentinib, detectable production of any infectious virus was delayed until 48 hours (L.O.D. = 5 TCID$_{50}$/mL). A time-of-addition study conducted in H1650 cells indicated that bemcentinib inhibition was most effective when present at early timepoints during SARS-CoV-2 infection, consistent with a role of AXL in virus entry (**Fig 5F**). Also in H1650 cells, the ability of bemcentinib to inhibit recently emerged SARS-CoV-2 variants of concern (VOC), Alpha (B.1.1.7) and Beta (B.1.351), was evaluated. While the Alpha VOC replicated poorly in these cells, bemcentinib significantly inhibited virus replication of both variants, providing evidence that the efficacy of the AXL inhibitor is not influenced by SARS-CoV-2 adaptative changes (**S5B Fig**).

SARS-CoV-2 infection of TMPRSS2$^{hi}$ Calu-3 cells (**S5A Fig**) was not sensitive to bemcentinib or E-64, providing evidence that in this cell line the route of virus entry was dominated by the TMPRSS2-dependent path, bypassing the use of PS receptors and the endosomal compartment (**Fig 5G**). Consistent with this, PS liposomes had no effect on SARS-CoV-2 viral loads in Calu-3 (**S5C Fig**). These findings stand in contrast to SARS-CoV-2 infection of TMPRSS2$^{+}$ H1650 cells that were markedly bemcentinib and E-64 sensitive and were found to be insensitive to camostat inhibition (**Figs 5B, S5A and S5D**). The paradoxical finding that virus entry into H1650 is sensitive to E64 and bemcentinib despite endogenous TMPRSS2 expression indicates that TMPRSS2-dependent pathways are not always the dominating or default route of SARS-CoV-2 entry and suggests that a more complex balance of events controls which pathway is used. Neither the total amount of cell surface expressed ACE2 nor the intracellular versus surface ACE2 ratio appears to determine the route of virus uptake (**S5E Fig**).

RNA sequencing studies confirmed and extended our findings with bemcentinib in A549$^{ACE2}$ cells. At 24 hpi, 20% of the transcripts in A549$^{ACE2}$ cells mapped to the viral genome. Infection in the presence of 1 μM bemcentinib significantly decreased the number of viral transcripts present (**Fig 5H**). Further analyses of potential qualitative changes in viral transcripts indicated that transcript numbers across the genome were reduced, rather than a reduction of specific subgenomic transcripts.

To directly evaluate the importance of AXL during infection of human lung cells, CRISPR-Cas9 technology was used to knock out (KO) AXL expression in H1650 and HCC2302 cells. H1650 AXL$^{neg}$, a biologically cloned AXL-null line, was evaluated along with bulk AXL KO populations of H1650 and HCC2302, denoted as AXL$^{low}$. The AXL$^{neg}$ clone, which expressed undetectable levels of AXL protein (**Fig 6A**), supported dramatically lower SARS-CoV-2 virus loads at a range of input MOIs (**Fig 6B**) and the low level of infection that was poorly inhibited by bemcentinib (**Fig 6C**). This reinforces that AXL plays an important role in SARS-CoV-2 infection of some lung cells and indicates that bemcentinib is poorly effective/ineffective in the absence of AXL, suggesting specificity. Bulk, uncloned populations of AXL$^{low}$ H1650 and HCC2302 (AXL$^{low}$) also supported reduced levels of SARS-CoV-2 infection and infection was poorly inhibited by bemcentinib (**S6 Fig**). Taken together, the data presented here demonstrate that SARS-CoV-2 utilizes AXL to enhance infection in some human lung cell lines, and that this mechanism can be effectively disrupted in by small molecule inhibitors or genetic ablation of AXL.

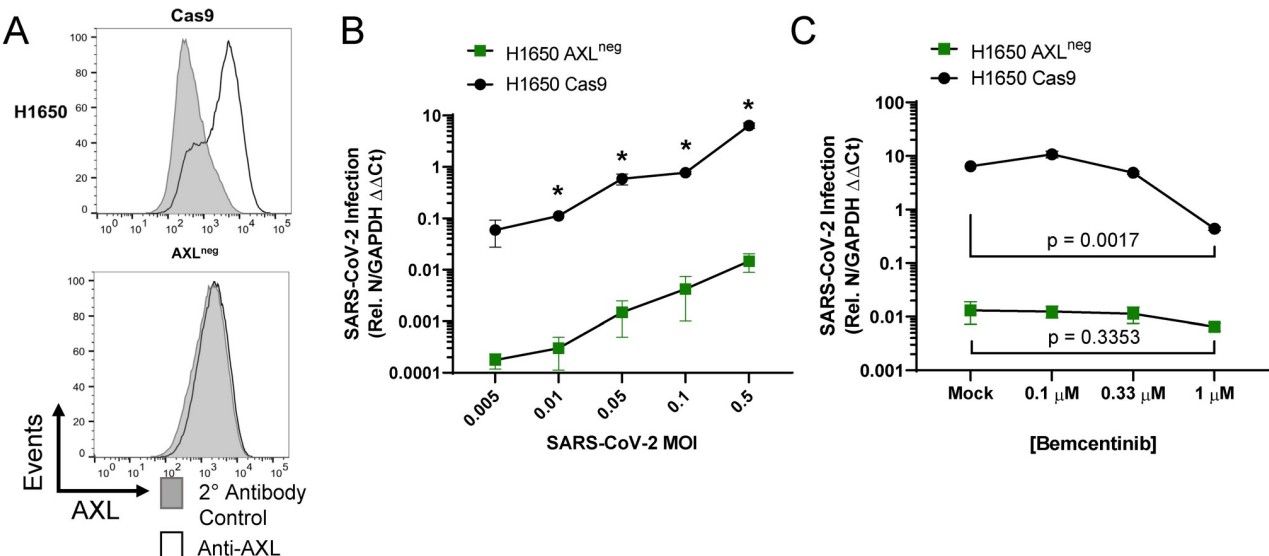

**Fig 6. AXL knockout reduces viral loads and ablates inhibition by bemcentinib. A)** Flow cytometry histograms depicting AXL surface staining (black) and secondary only background (gray) on parental, Cas9 expressing H1650 cells or Cas9/guide RNA treated and cloned cells, demonstrating loss of AXL expression. H1650 AXL knockout cells were generated by lentiviral transduction of Cas9 and gRNA targeting AXL, followed by selection, enrichment, and biological cloning. **B)** H1650 AXL[neg] and H1650 Cas9 (parental) lines were challenged with SARS-CoV-2 at indicated MOIs. At 24hpi, viral loads were assessed by RT-qPCR. **C)** H1650 parental and AXL[neg] lines were treated with the indicated concentration of bemcentinib and challenged with SARS-CoV-2 (MOI = 0.5) and viral loads determine by RT-qPCR 24 hpi. Data are pooled from 3 independent experiments (**B**, **C**) or are representative of at least 3 experiments (**A**). Data represented as means ± SEM. Multiple t-tests; asterisks represent $p < 0.05$ (**B**). Student's t-tests comparing Mock and 1μM; p values shown (**C**).

## AXL facilitates infection of other betacoronaviruses

To assess the role of AXL in the related betacoronavirus, mouse hepatitis virus (MHV strain A59), we investigated the ability of bemcentinib to inhibit infection in C57BL/6J mouse bone marrow derived macrophages (BMDMs) that express AXL [34–36]. As MHV uses the mouse receptor CEACAM as its cognate receptor, this model allowed us to examine the role of AXL with a coronavirus that is ACE2-independent [37]. Bemcentinib added to BMDM cultures decreased virus load at 24 hours in a dose-dependent and an MOI-dependent manner (S7A and S7B Fig). At higher MOIs, the effect of bemcentinib was abolished. Bemcentinib treatment also inhibited MHV infection in peritoneal macrophages, an MHV permissive population phenotypically and functionally distinct from BMDMs (S7C Fig). To complement pharmacological inhibition studies, we cultured peritoneal macrophages from mice with a genetically disrupted AXL, AXL[lacz]. MHV infection of peritoneal macrophages from these animals was significantly suppressed relative to WT controls (S7D Fig). Finally, we leveraged this robust set of tools for examining PS receptor utilization during infection to examine potential interactions during SARS-CoV entry. SARS-CoV spike-bearing pseudovirion entry in Vero E6 cells was significantly inhibited by bemcentinib (relative to Untreated) and PS liposomes (relative to PC liposomes) (S7E Fig). These studies provide evidence that AXL facilitates infection of multiple members of this enveloped virus family, independent of the cognate receptor used by the virus.

## Discussion

Here, we demonstrate that PS receptors, AXL, TIM-1 and TIM-4, potentiate SARS-CoV-2 infection of HEK 293T cells when the cognate receptor for the virus, ACE2, was expressed at

low levels. PS receptors enhanced virion binding to cells in a PS-dependent manner. At higher levels of ACE2 expression, a role for the PS receptors was no longer observed. Similar findings were observed for TMPRSS2-facilitated, ACE2-dependent infection, indicating that when ACE2 is expressed on the plasma membrane at high concentrations these host proteins that assist SARS-CoV-2 entry are no longer required.

A recent study reported that AXL mediates SARS-CoV-2 infection [26]. That report suggested that AXL-mediated virus entry is independent of ACE2 and that AXL binds to the N-terminal domain of SARS-CoV-2 spike. Conclusions from our studies indicate that AXL and other PS receptors mediate enhancement of SARS-CoV-2 infection through interactions with virion-associated PS and enhance SARS-CoV-2 in an ACE2-dependent manner. We report several lines of evidence that are consistent with our contention that the PS receptors interact with virion-associated PS. First, PS is readily detectable on the surface of SARS-CoV-2 virions. Second, PS liposomes abrogate binding and entry in a dose-dependent manner. Third, interfering with PS/PS receptor complexes by mutating the TIM-1 PS binding pocket abrogates SARS-CoV-2 pseudovirus infection, and reduces virion internalization. These data are consistent with and support the well-established mechanism of PS receptor enhancement of enveloped virus infection [11–13,38,39]. Fourth, we directly tested whether AXL binds to purified spike or NTD by flow cytometry and biolayer interferometry assays and were unable to detect any interaction. Finally, the ability of diverse PS receptors to enhance SARS-CoV-2 infection in HEK 293T cells lends support to the contention that these receptors interact with PS rather than viral spike on the surface of SARS-CoV-2 to mediate productive infection. Thus, we conclude that AXL does not interact with SARS-CoV-2 spike, nor does it mediate virus entry unilaterally. It should be noted that this is the first example of an enveloped virus that utilizes PS receptors in conjunction with low/moderate expression of a high affinity surface receptor.

PS receptors have previously been shown to interact with PS on the surface of other enveloped viruses such as filoviruses, alphaviruses and flaviviruses and mediate internalization into endosomes [11,40]. However, PS receptor-dependent entry is a mechanism that is functionally out-competed by high-affinity viral glycoprotein-host receptor interactions, such as that of Lassa virus with α-dystroglycan [16,19]. In the case of Lassa virus entry PS receptors seem to serve as a backup entry mechanism, as these receptors only mediated virus internalization when the high affinity surface receptor, α-dystroglycan, was not expressed. Our ACE2 dose response studies in HEK 293T cells are consistent with this, where PS receptors only facilitate virus infection when ACE2 is limiting.

To this point, a previous study concluded that the related coronavirus, SARS-CoV, was not productively internalized by PS receptors [11]. However, with insights from our studies, an alternative explanation is that PS receptor enhancement of coronavirus entry is ACE2-dependent and sufficient quantities of ACE2 on the plasma membrane abrogate a role for PS receptors. Thus, PS receptors only facilitate SARS-CoV-2 entry under conditions where ACE2 is expressed at suboptimal levels, conditions that were likely not evaluated in the cited study but are found on ACE2 expressing cells such as Vero E6 cells and many patient-derived lung cell lines. Consisted with our SARS-CoV-2 data, SARS-CoV pseudovirus experiments in a human lung cell line support these hypotheses. As ACE2 expression is low within the lung, such suboptimal conditions may be highly relevant during coronavirus infections [41,42].

The preferential utilization of AXL rather than TIM-1 by SARS-CoV-2 in Vero E6 cells was unexpected. In our previously studies, other enveloped viruses that utilize PS receptors, such as filoviruses, use TIM-1 preferentially when both proteins are expressed [12,19]. Further, a recent study identified that the TIM-1 IgV domain that contains the PS binding pocket serves as an effective inhibitor of enveloped virus infection regardless of the PS receptor utilized for virus uptake [43], consistent with the good affinity the TIM-1 PS binding pocket has for PS

[44]. Nonetheless, when TIM-1 is not present in cells and AXL is the sole PS receptor expressed, AXL is used by filoviruses and flaviviruses [11,13,45–47]. The subpar utilization of AXL reported for other viruses may be due to the requirement for the adaptor protein, Gas6, to also be present. Alternatively, steric factors on the virion may control which receptor is preferred. As multiple proteins from a variety of different PS receptors families can mediate uptake of apoptotic bodies, it is no surprise that PS receptor interactions with viruses are likewise intricate. Further studies are needed to understand the mechanism driving preferential use of AXL by SARS-CoV-2.

Our studies provide evidence that PS receptors enhance SARS-CoV-2 binding to cells and mediate internalization into endosomes where cysteine proteases potentiate spike protein triggering and subsequent fusion events. Consistent with the utilization of this uptake pathway, the cysteine protease inhibitor E-64 effectively blocked ACE2 or ACE2/PS receptor entry in HEK 293T cells. This is also supported by our super resolution microscopy observations in Vero E6 cells showing that AXL and ACE2 colocalize within the cell during SARS-CoV-2 infection. However, in HEK 293T, the route of virus entry changes upon expression of TMPRSS2 and virion entry is no longer sensitive to E-64. As others have reported, these findings are consistent with TMPRSS2-dependent entry dominating as the route of entry when TMPRSS2 is expressed [23,48,49]. We also investigated virus infection of a variety of lung lines that endogenously express TMPRSS2, AXL, and ACE2. While findings with Calu-3 cells were similar to that we observed in TMPRSS2-transfected HEK 293T cells, the other TMPRSS2$^+$ lung lines, such as H1650 cells, remained sensitive to E-64 and the AXL signaling inhibitor, bemcentinib. Differences in the ability of TMPRSS2 expression to control the route of entry may be due to a fine balance of surface expression of the various receptors and should be explored in more detail in future studies.

Our data indicate that AXL serves as the most important PS receptor for SARS-CoV-2 infection of the TIM and TAM families and our studies with MHV implicated AXL in facilitating infection of additional coronaviruses. While AXL is abundant on lung epithelial cells, it is also present in many organs in the body, with the exclusion of neural tissues [33,50]. Thus, it is likely a role for AXL in SARS-CoV-2 infection is not only relevant to lung cell populations, but ACE2-expressing tissues suspected to be affected by COVID-19 such as the heart and kidneys [51,52]. We surmise that AXL-inhibiting therapeutics could function in tandem with other antivirals, protecting a number of organs from infection. Our data suggest that the efficacy of bemcentinib will persist as the virus evolves, inhibiting the VOCs Alpha and Beta effectively. By targeting host proteins such as AXL we dramatically reduce the potential selection for pathogen mutants that reduce or abolish antiviral activity. Given that currently utilized small molecule therapeutics such as remdesivir targeting viral proteins have shown limited efficacy and the benefits of antibody-rich convalescent plasma is minimal, AXL inhibition by small molecule inhibitors such as bemcentinib offers a novel route of attack to reduce SARS-CoV-2 entry and disease [53,54].

The specificity of bemcentinib for AXL is well documented, and compared to alternative inhibitors, such as BMS-777607, is relatively high [55]. However, it must be noted that AXL is a multifunctional protein with roles in promoting cell growth, cell migration, tumor invasion, and dampening inflammatory cytokine production [55]. While our results indicate a consistent role for AXL in SARS-CoV-2 entry, we cannot exclude the possibility that these alternative facets of AXL activity, when inhibited, also reduce SARS-CoV-2 infection. The most pertinent to viral infection would be the effect of AXL on inhibiting type I interferon responses [56,57]. Bemcentinib treatment has been shown to remove this suppression and increase IFN-beta mRNA during ZIKV infection [58]. We have conducted studies examining this possibility as well, and have observed both enhancement and suppression of specific ISGs, but the relevance

of these changes is unclear for SARS-CoV-2 which can antagonize IFN responses via Orf6 [59,60].

Bemcentinib is currently in Phase II trials for non-small cell lung cancer (NSCLC) and a variety solid and hematological cancers (ClinicalTrials.gov IDs: NCT03184571, NCT03184558). However, multiple screens have identified bemcentinib as inhibitory to SARS-CoV-2 infection, bolstering this mechanism of entry [61,62]. Two phase 2 clinicals trial evaluating efficacy of bemcentinib in hospitalized COVID-19 patients are ongoing, with the first report suggesting short-term efficacy (https://clinicaltrials.gov/ct2/show/NCT04890509). In this exploratory, open-label study bemcentinib was added to standard-of care (SoC) therapy to hospitalized patients [63]. Though the primary endpoints (time to improvement by 2 points on WHO ordinal scale or time to discharge) showed a marginal benefit of bemcentinib treatment, there was evidence of potentially meaningful clinical benefit in a key secondary endpoint which was avoidance of deterioration. Given the *in vitro* data shown in our work, the key therapeutic window for bemcentinib may be prophylactic rather than post-hospitalization.

The robust body of PS receptor research completed in the last decade and historical patterns of zoonotic events (Ebola virus, Zika virus, SARS-CoV) suggest that future emergent viral pathogens are likely to utilize PS receptors to enhance entry and infection [64]. The observations reported here that PS receptors are utilized by a novel pandemic coronavirus support this conclusion. This confluence of information provides insights into a new class of potential therapeutics to stem future outbreaks, namely drugs aimed at inhibiting PS receptor activity. Further studies are required to determine the role of TIM and TAM use *in vivo*; however, our studies demonstrate a role of PS receptors in SARS-CoV-2 infection in relevant cell populations and further extend the importance of PS receptors in enveloped virus entry to coronaviruses.

## Materials & methods

### Ethics statement

This study was conducted in strict accordance with the Animal Welfare Act and the recommendations in the Guide for the Care and Use of Laboratory Animals of the National Institutes of Health (University of Iowa (UI) Institutional Assurance Number: #A3021-01). All animal procedures were approved by the UI Institutional Animal Care and Use Committee (IACUC) which oversees the administration of the IACUC protocols and the study was performed in accordance with the IACUC guidelines (Protocol #8011280, Filovirus glycoprotein/cellular protein interactions).

### Mice

The mice (6–8 weeks old, female) used for macrophage isolation in these studies were obtained from the Jackson Laboratory (C57BL6/J and AXL$^{lacz}$). The protocol (#8011280) was approved by the Institutional Animal Care and Use Committee at the University of Iowa.

### Primary cells and immortal cell lines

Bone marrow derived macrophages (BMDM) and peritoneal macrophages were isolated and cultured in RPMI-1640 supplemented with 10% Fetal Bovine Serum (FBS), 0.5 μg/mL of penicillin and streptomycin (pen/strep) and 50 ng/mL murine M-CSF. BMDMs were matured for 5 days, and peritoneal macrophages for 2 days. Vero E6 cells (ATCC; CRL-1586), Vero TMPRSS2, Vero E6 and HEK 293T (ATCC; CRL-11268) were cultured in Dulbecco's modified Eagle's medium (DMEM, GIBCO, Grand Island, NY) supplemented with 5–10% FBS and 1%

penicillin/streptomycin (GIBCO). Blasticidin (5 μg/mL, Invivogen; ant-bl-05) was added to media supporting Vero E6 TMPRSS2 cell growth. A549$^{ACE2}$ cells were generated by transduction of A549 (ATCC; CCL-185) with a codon-optimized ACE2 encoding lentivirus and selection with 10 μg/mL blasticidin. Clonal populations were isolated and ACE2 expression verified by western blot. H1650, HCC2302, and HCC1944 human lung lines were maintained in RPMI with 5–10% FBS and pen/strep. Cell lines were periodically tested for mycoplasma contamination (Bulldog Bio; 25233) and cured of contamination before use (Plasmocin, Invivogen; ant-mpp). Cell lines were authenticated periodically by ATCC (A549 and HEK 293T) or the lab responsible for their generation (H1650, HCC2302, HCC1944, and Vero TMPRSS2).

AXL knockout HCC2302 and H1650 were generated by transduction of parental cells with a Cas9 encoding lentivirus (kind gift of Aloysius Klingelhutz, University of Iowa) and selection in 10 μg/mL blasticidin for 10 days. Then cells were transduced with an Invitrogen LentiArray CRISPR gRNA lentivirus targeting AXL (Thermo), and subsequently selected in puromycin at 2μg/mL for 5 days. Cells were then lifted, stained for AXL, and sorted for AXL$^{low}$ cells at the University of Iowa Flow Cytometry Core on a FACSAria Fusion (Becton, Dickinson and Company, Franklin Lakes, New Jersey). Bulk populations of AXL$^{low}$ cells were used for experiments as noted, and Clone #4 was generated by sorting single AXL$^{low}$ cells into a 96 well plate. AXL expression was verified by flow cytometry on a FACSVerse (Becton, Dickinson and Company).

## Viruses

Studies used the 2019n-CoV/USA-WA-1/2020 strain of SARS-CoV-2 (BEI; MT985325.1) which was propagated on Vero TMPRSS2 cells. Briefly, Vero TMPRSS2 cells were inoculated with an MOI of 0.001 in DMEM supplemented with 2% FBS and pen/strep. Media was removed and refreshed 24 hpi. When cells exhibited severe cytopathic effect, generally 72 hpi, cells were freeze-thawed once, transferred to a conical tube, centrifuged at 1000g for 10 minutes, and supernatants were filtered through a 0.45 μm filter. Virus was sequenced via Sanger sequencing periodically for furin cleavage site mutations (none were detected) and only low passage stocks were used.

MHV (A59) stocks were generously provided by Dr. Stanley Perlman. Viral stocks were generated on Vero E6 and the TCID$_{50}$ was determined on HeLa-mCECAM1 cells by identification of cytopathic effect at 5 days.

Stocks of recombinant vesicular stomatitis virus that expressed SARS-CoV-2 spike containing the D614G mutation and nanoluciferase (rVSV/Spike) (kind gift of Dr. Melinda Brindley, Univ. GA) were generated in either Vero E6 or Vero E6 TMPRSS2 cells. Cells were infected with a low MOI (~0.005) of virus and input was removed after ~12 h. Upon evidence of cytopathology, supernatants were collected over a three-day period, filtered through a 0.45 μm filter and frozen at -80˚C until purified. Supernatants were thawed and centrifuged overnight at 7000 x g to concentrate the virus. The virus pellet was resuspended in endotoxin-free PBS and layered over a 20% sucrose/PBS cushion. Virus was pelleted through the cushion by centrifugation at 28,000 rpm in a SW60Ti rotor (Beckman). The virus pellet was resuspended in PBS and the TCID$_{50}$ was determined on Vero TMPRSS2 cells.

FITC-labeled rVSV/spike was generated using FITC 'Isomer I' (Thermo; F1906). Briefly, concentrated VSV/spike was resuspended in pH 9 carbonate buffer, to which 1 μL FITC in DMSO was added (per 100 μL virus). This labeling reaction was conducted for 1 hour on ice, after which the virus was injected into a hydrated Slide-A-Lyzer dialysis cassette (Thermo; 66383). Sample was dialyzed for 6 hours in 3L PBS at 4 degrees C in the dark, with dialyte changes every 2 hours. Virus was then sucrose cushion purified and used in internalization

assay. FITC labeling did not significantly change the infectious titer of the stock, relative to an unlabeled control.

All viral titers were determined by a modified Spearman-Karber method as previously described and reported as infectious units (IU)/mL [65].

## Inhibitors and liposomes

Bemcentinib (BGB324, R428) was provided by BerGenBio ASA (Bergen, Norway) and dissolved in DMSO for *in-vitro* studies. BMS-777607 (Millipore Sigma; BM0019) was dissolved in DMSO. E-64 (Millipore Sigma; 324890) was dissolved in DMSO. PS and PC liposomes were prepared as previously described [25]. Briefly, lipids were dissolved in chloroform, dried, resuspended in PBS, and sonicated for 5 minutes on ice. Liposomes were aliquoted and stored at -80˚C until use. Inhibitors and liposomes were added to cells 1 hour before infection/transduction/binding, unless otherwise noted.

## RNA isolation and qRT PCR

Total RNA for PCR was extracted from cells or tissue using TRIzol (Invitrogen; NCC1701D) according to the manufacturer's protocol. Total isolated RNA (1 μg) was reverse transcribed to cDNA using the High-Capacity cDNA Reverse Transcription Kit (Applied Biosystems; 4368814). The resulting cDNA was used for amplification of selected genes by real-time quantitative PCR using Power SYBR Green Master Mix (Applied Biosystems; 4368708). Data were collected on QuantStudio 3 and Ct values determined with the QuantStudio Data Analysis software (Applied Biosystems). Averages from duplicate wells for each gene were used to calculate relative abundance of transcripts relative to housekeeping genes (GAPDH, mouse cyclophilin) and presented as $2^{-\Delta\Delta CT}$. Primers used herein (Integrated DNA Technologies) are listed below, 5' to 3' in format.

GAPDH$^{for}$: GGT GTG AAC CAT GAG AAG TAT GA
GAPDH$^{rev}$: GAG TCC TTC CAC GAT ACC AAA G
CypA$^{for}$: GCT GGA CCA AAC ACA AAC GG
CypA$^{rev}$: ATG CTT GCC ATC CAG CCA TT
SARS-CoV-2 N$^{for}$: CAA TGC AAT CGT GCT AC
SARS-CoV-2 N$^{rev}$: GTT GCG ACT ACG TGA TGA GG
SARS-CoV-2 S$^{for}$: CTA CAT GCA CCA GCA ACT GT
SARS-CoV-2 S$^{rev}$: CAC CTG TGC CTG TTA AAC CA
MHV nsp12$^{for}$: AGG GAG TTT GAC CTT GTT CAG
MHV nsp12$^{rev}$: ATA ATG CACCTG TCA TCC TCG
VSV M$^{for}$: CCT GGA TTC TAT CAG CTT C
VSV M$^{rev}$: TTG TTC GAG AGG CTG GAA TTA

## VSV/Spike pseudovirus production

The production of SARS-CoV-2-Spike vesicular stomatitis virus (VSV/Spike) pseudovirions has been described previously [3]. Briefly, HEK 293T cells were seeded in 10 cm tissue culture plates (CellTreat; 229692). After 48 hours cells were transiently transfected with a SARS-CoV-2-Spike pCG1 plasmid (a kind gift from Dr. Stefan Pohlman as described [3]) using a standard polyethyleneimine (PEI) protocol. For this transfection, one tube was prepared with 16 μg of plasmid diluted in 1.5 mL of OPTI-Mem (GIBCO; 31985–070). The second tube was prepared with PEI (1 mg/mL) diluted in 1.5 mL of OPTI-Mem at a concentration of 3 μl/1 μg of DNA transfected. The tubes were then combined, vortexed for 10–15 seconds and left to incubate at room temperature for 15 minutes. The mixture was then added dropwise to the HEK 293T

cells and returned to incubator overnight. Twenty-four hours after transfection the cells were infected with a stock of replication insufficient VSV virions expressing firefly luciferase that were pseudotyped with Lassa virus glycoprotein on the viral membrane surface. The infection was incubated for ~6 hours at 37°C, was removed from the cells, and fresh media was added. Viral collection took place at 24- and 48-hours post-infection. Media supernatants were removed from the flasks, briefly spun down to remove cellular debris (180 x g for 1 minute) and filtered through a 0.45 μm syringe-tip disk PVDF filter (CellTreat; 229745). Supernatants were then concentrated by a 16-hour centrifugation step at 5380 x g at 4°C. Pseudovirions were purified through a 20% sucrose cushion via ultracentrifugation at 28,000 rpm for two hours at 10°C in a Beckman Coulter SW60Ti rotor. Pseudovirus was then resuspended in 1x PBS and stored at -80°C. Pseudovirus was titered using end point dilution on Vero E6 cells. All infections of cells using SARS-CoV-2-S pseudotyped virions were conducted at a volume of virus that gave a relative light unit (RLU) of roughly 100,000–200,000 RLU.

## HEK 293T transfections and plasmids

All transfections were performed in HEK 293T cells, with a total plasmid concentration of 2 μg. Cells were seeded into 6-well tissue culture plate (Dot Scientific; 667106) at a density of 5 x $10^5$ cells/well. Forty-eight hours after cell seeding, cells were transfected with CMV-driven expression vectors of ACE2, TIM-1, ND115DN TIM-1, TIM-4, AXL, TYRO3, MerTK and TMPRSS2 (see plasmid details below) with a standard PEI transfection protocol. For this transfection one tube was prepared with plasmid DNA and 150 mM NaCl at a concentration of 25 μl/1 μg of DNA transfected. A second tube was prepared with 150 mM NaCl at a concentration of 25 μl/1 μg of DNA transfection along with PEI (1 mg/mL) at a concentration of 3 μl/1 μg of DNA transfected. Tubes were combined, vortexed vigorously for 10–15 seconds and incubated at room temperature for 15 minutes. Mixtures were added dropwise to HEK 293T cells. For all experiments using ACE2 50 ng of ACE2 plasmid were transfected, unless otherwise noted. For PS receptors, 1000ng of plasmid was transfected unless otherwise noted. All transfections were brought up to 2 μg total transfected DNA with a PCD3.1 empty expression vector.

## ACE2 and PS receptor expression detection via flow cytometry

To detect cell surface expression of ACE and PS receptors on transfected HEK293Ts, WT Vero E6s, and AXL knock down/out H1650 and HCC2302 clones, cells were lifted using 1x Versene (GIBCO; 15040066) at 37°C for ~15 minutes and placed in 5mL polystyrene round-bottom tubes (Falcon; 352052). Cells were washed once with FACS buffer (1x Sterile PBS, 2% FBS, 0.05% sodium azide). Cells were incubated for 30 minutes on ice with primary antibodies diluted in FACS buffer against ACE or PS receptors. Primary antibodies were diluted to 0.75 μg/mL in FACS buffer prior to incubation. Specific primary antibodies used as follows: goat anti-ACE2 (R&D; AF933), goat anti-AXL (R&D; 154), goat anti-TIM-1 (R&D; 1750), goat anti- Tyro3 (R&D; AF859), goat anti-TIM-4 (R&D; 2929), goat anti-MerTK (R&D; AF891), rabbit anti-TMPRSS2 (ABCAM; ab92323). Cells were washed once with FACS buffer. Cells were then incubated with secondary antibodies at a 1:1000 dilution in FACS buffer on ice for 30 minutes. Secondaries used were donkey anti-goat IgG (H+L) Alexa Fluor 647 (Jackson Immuno Research; 705-605-003) and donkey anti-rabbit IgG (H+L) Alexa Fluor 647 (Invitrogen; A32733). Flowcytometry was performed on a Becton Dickinson FACS Calibur and analyzed by Flow Jo.

To examine both the surface and intracellular expression of hACE2 on H1650, Calu-3, HCC1944, HCC2302, A549$^{ACE2}$ and Vero E6 cells we performed the following protocol.

Briefly, cells were staining with Fixable Viability Dye eFluor 780 (eBioscience; 65-0865-14), goat anti-human ACE2 (R&D; AF933) followed by secondary were donkey anti-goat IgG (H +L) Alexa Fluor 647 (Jackson Immuno Research; 705-605-003) To measure the intracellular expression of hACE2, cells were surface stained with Fixable Viability Dye eFluor 780, fixed (PFA 4%), permeabilized (1X PBS + 0.5%Tween20) and stained intracellularly using goat anti-human ACE2 (R&D; AF933) followed by secondary donkey anti-goat-AF647. Unstained cells, cells plus viability dye and cells plus secondary antibody/viability dye were included as a control in every staining. Samples were measured on a FACSverse cytometer (BD Biosciences) and data were analyzed with Flowjo software (BD Biosciences).

## VSV/SARS-CoV-2-spike pseudovirion studies with inhibitors

Following the transfection of HEK 293T cells, cells were incubated for 24 hours. At that time cells were lifted with 0.25% Trypsin (GIBCO; 25200–056) and plated at a density of 2 X $10^4$ cells/well on opaque, flat-bottomed, 96-well plates (Falcon; 353296). Each transfection was plated into at least 3 wells to create experimental replicates. Cells were incubated for an additional 24 hours. At that time, cells were infected with VSV-luciferase/SARS-CoV-2 spike. Cells were incubated for an additional 24 hours. For experiments done with inhibitors, cells were treated with the concentrations of inhibitors noted in the figure panel immediately prior to being infected with pseudotyped virions. After 24 h, virus-containing media was removed and replaced with 35 μL of 1x Passive lysis buffer (Promega; E194A). Plates underwent three freeze-thaw cycles consisting of freezing on dry-ice for 15 minutes followed by thawing at 37°C for 15 minutes. We followed the protocol for measuring firefly luciferase as reported previously (Johnson et al., 2017). For this method 100 μL of luciferin buffer (100 μl of luciferin buffer (15 mM MgSO$_4$, 15mM KPO$_4$ pH 7.8, 1 mM ATP, and 1mM dithiothreitol) and 50 μL of 1mM d-luciferin potassium salt (Syd Laboratories; MB000102-R70170)) were added to each well and luminescence was read via Synergy H1 Hybrid reader (BioTek Instruments). Relative luminescence units were read out. Results were analyzed by normalizing values to mock transfection with no protease inhibitors.

## HEK 293T SARS-CoV-2 infection studies

Following PEI transfection with plasmids as described in the previous section, cells were lifted with 0.25% trypsin and plated into 48-well plates at a density of 6 x $10^4$ (Dot Scientific; 667148). In our BSL3 facility, transfected HEK 293T cells were infected at a MOI = 0.5 with SARS-CoV-2. Cells were incubated at 37°C or 24 hours and then treated with TRIzol RNA isolation reagent and removed from the BSL3 facility. RNA extraction and cDNA generation proceeded as described. RT-qPCR was conducted on the cDNA using SARS-CoV-2-Spike and GAPDH primers. Data analyzed using the ΔΔCt method as described above.

## Virion binding assays

For Vero E6 binding studies, cells were grown to confluence in 48 well plates. Media was replaced with DMEM supplemented with 10% FBS and the indicated compounds. Cells were incubated at 10°C (preventing internalization and entry) until equilibrated and SARS-CoV-2 was added at MOI 5. Plates were returned to 10°C for 1 hour. Media was removed and cells were washed three times with cold DPBS (GIBCO; 14190144), removing any unbound virus. Then 0.05% Trypsin-EDTA (GIBCO; 25300054) was added to control wells for 5 minutes at 37°C, and washed. After the final wash, all media was removed and replaced with TRIzol. RNA was isolated and analyzed as described.

For HEK 293T binding studies, cells were transfected as described and left to rest for 48 hours. Cells were lifted with Versene (GIBCO), media was replaced with DMEM supplemented with 10% FBS, and cells were plated in IMMULON 2HB flat bottom plates (Thermo Scientific, Waltham, MA). Cells were incubated at room temperature (preventing internalization and entry) until equilibrated and rVSV-SARS-CoV-2 spike virions (rVSV/Spike) were added at MOI 5. Plates remained at room temperature 1 hour. Media was then removed and cells were washed three times with cold DPBS via repeated centrifugation (300 RCF) and decanting steps, removing any unbound virus. After the final wash, all media was removed and replaced with TRIzol. RNA was isolated and analyzed as described.

## Phosphatidylserine ELISA

Sucrose cushion purified SARS-CoV-2 (UV irradiated, 30 minutes), rVSV/spike, and Adenovirus (empty vector) were coated on Immulon 2HB plates in pH 9 binding buffer. Virus amount was normalized to protein content, as determined by Bradford assay (BioRad; 5000006), and coating was done at 4 degrees C overnight. Plates were washed three times with PBS + 0.02% Tween-20, and subsequently blocked with PBS + 2% bovine albumin fraction V (Fisher; 9048-46-8) overnight at 4 degrees C. Plates were again washed and incubated for 2 hours at room temperature with the antibody Bavituximab (generous gift of Rolf Brekken) diluted 1:200 in PBS + 2% FBS. After three washes the secondary antibody, HRP conjugated goat anti-human IgG Fc (Invitrogen; A18817) for 2 hours at room temperature. Plates were developed using TMB substrate (BD; 555214) and stopped with 2N sulfuric acid. Absorbance was detected using an Infinite M200 Pro (Tecan) plate reader and analyzed in GraphPad.

## HEK 293T rVSV/Spike internalization

HEK 293T were transfected as described, using 1 μg of each plasmid. 24 hours after transfection, cells were lifted in Versene (GIBCO) and 1E5 cells plated in each well of a 96-well V-bottom plates (Brand GMBH; 781661). The plate was kept on ice until equilibrated, and FITC labeled rVSV-spike was added (0.33 μL/well). Virus was allowed to bind for 1 hour on ice, and well were then washed 3 times in RPMI 1640 + 10% FBS. Bemcentinib was added to indicated wells after resuspending cells in RPMI + 10% FBS, and the plate was shifted to 37 degrees C for 30 minutes. After this time for internalization elapsed, cells were pelleted, washed once, and resuspended in 0.25% Trypsin-EDTA (GIBCO) to remove all non-internalized virions. Binding controls were not treated with trypsin, and the bind-tryp controls were trypsin treated before the internalization step. After 10 minutes at 37 degrees C, trypsin was neutralized and cells were washed 3 times with FACS buffer (PBS + 2% FBS + 0.05% azide). Internalized virus was then quantified by FITC gMFI as measured by flow cytometry (FACSCalibur, BD), and values were normalized to empty vector transfected internalization controls.

## RNA sequencing and analysis

Following indicated treatments and infections that were performed in triplicate or quadruplicate, Vero E6 and A549[ACE2] cells in 6 well formats were homogenized using QIAShredder tubes and total RNA isolated using the RNEasy kit with DNase treatment (Qiagen; 74004 and 79254). High quality RNA samples that were verified by Bioanalyzer (Agilent) was quantified and used as input to generate mRNA-seq libraries for the Illumina platform. Paired-end sequencing reads were subject to alignment to suitable reference genomes: human GRCh38 (GCA_000001405.15—A549 cells), green monkey (Chlsab1/GCA_000409795.2—Vero E6 cells) and SARS-CoV-2 (MN985325—both A549 and Vero E6 cells). Alignments to human and monkey genomes were performed using hisat2 v2.0.5, while to viral genome using bowtie2

v2.2.9. Aligned reads were counted using feature Counts from subread package v1.5.2. Counted reads were normalized in R, using DESeq2 v1.30.0 and subjected to statistical analysis. The statistical analysis included computation of median based fold changes, Student t-test p values and false discovery rate (multiple testing correction).

### Purified spike protein flow cytometry binding studies

The NTD-Fc and RBD-Fc constructs were kindly provided by Tom Gallagher. They contain the Fc region of human IgG1 fused to the N-terminal domain of SARS-CoV-2 spike (residues 1–309) or the RBD-containing C-terminal domain of the S1 subunit (residues 310–529). We also generated an Fc-Spike construct that contains the Fc region of human IgG1 fused to a cleavage-negative form of the Spike ectodomain (subunits S1 and S2, corresponding to positions 1–1274). To eliminate the polybasic furin cleavage site of spike, we replaced the Arg-Arg-Ala-Arg motif at positions 682–685 with Ser-Ser-Ala-Ser. All proteins were produced by transient transfection of 293T cells using PEI. Proteins were harvested in 293S ProCDM and purified using Protein A beads. Eluted products were dialyzed against phosphate buffered saline pH 7.4. All proteins were analyzed by SDS-PAGE and gels were silver stained to verify the purity of the eluted product.

We measured the binding efficiency of anti-NTD antibody AM121 (Acro Biosystems) to the Spike-based constructs using ELISA, as previously described [66,67]. For this purpose, the NTD-Fc, RBD-Fc or spike-Fc suspended in PBS were attached to 96-well protein-binding plates by incubation at isomolar concentrations (2, 1.37 and 5 µg/mL of the probes, respectively). The next day, wells were washed once with buffer containing 140 mM NaCl, 1.8 mM CaCl2, 1 mM MgCl2, 25 mM Tris pH 7.5, 20 mg/mL BSA and 1.1% low-fat milk. The anti-NTD antibody suspended in the same buffer was then added to the wells at 0.5 µg/mL. Binding of the anti-NTD antibody was detected using a goat anti-human kappa light chain conjugated to horseradish peroxidase (HRP) (BioRad; STAR127). To normalize for the amount of the bound probes, we also quantified the amount of probe bound to the wells by incubation with an HRP-conjugated goat anti-human antibody preparation. Binding of the HRP-conjugated antibodies was measured by luminescence using SuperSignal West Pico enhanced chemiluminescence reagents and a Synergy H1 microplate reader, as previously described [68].

To determine binding of the above probes to AXL, we used flow cytometry. Briefly, HEK 293T cells were seeded in 6-well plates (8.5E5 cells per well) and transfected the next day with 1.5 µg of empty vector or plasmids that express AXL or the full-length form of human ACE2 using JetPrime transfection reagent (PolyPlus). Three days after transfection, cells were detached using PBS supplemented with 7.5 mM EDTA and washed once with washing buffer (PBS supplemented with 5% newborn calf serum). Cells were then incubated with the NTD-Fc, RBD-Fc or Spike-Fc probes (at 5 µg/mL) or anti-AXL antibody (at 0.75 µg/mL) in the same buffer for one hour on ice and were washed four times with washing buffer. To detect binding of the Fc probes, we used a goat anti-human polyclonal antibody-Alexa 647. To detect binding of the anti-AXL antibody, we used a goat anti-donkey polyclonal antibody preparation conjugated to Alexa 594. Secondary antibodies were added at a 1:500 dilution and incubated with the cells on ice for one hour. Cells were then washed and analyzed by flow cytometry. Staining was evaluated on a FACS Calibur and analyzed by Flow Jo (BD).

### Biolayer interferometry

Biolayer interferometry was performed on an OctetRed96 (Pall Forte-Bio, USA) using NiNTA- (Forte-Bio; 185101) or Streptavidin- (Forte-Bio; 18–5019) coated Dip and Read biosensors for immobilisation of His-tagged or biotin-tagged proteins respectively. All samples

were diluted in Kinetic Buffer (0.1% BSA, 0.02% Tween 20, 0.05% Sodium azide in PBS). Biosensors were equilibrated in Kinetic Buffer, and protein (His-tagged SARS-Cov-2 S1 protein NTD, Acro Biosystems; S1D-C52H6; irrelevant control His-tagged β4 integrin fibronectin type III domain, generous gift from Petri Kursala laboratory; biotinylated tilvestamab anti-AXL antibody, BerGenBio, Norway) at a concentration of 0.7μM was loaded for 10 minutes. A baseline was taken for 2 minutes in Kinetic Buffer before testing association of the target binding protein for 10 minutes (AXL extracellular domain fused to human IgG Fc region, BerGen-Bio ASA, 2μM; positive control Anti-SARS-Cov2 spike NTD Neutralizing antibody, Acro Biosystems; AM121, 0.1μM; IgG1 isotype control antibody, BioXcell; BE0297, 0.1μM). Results were analyzed in Prism 9.1.1 for MacOS (GraphPad Software) by aligning to the baseline values immediately prior to association and subtracting signal from isotype control.

## STED sample preparation and image acquisition

12mm #1.5 coverslips (Fisher Scientific; 12-545-81P) were coated with 0.1% bovine Achilles' tendon collagen diluted in 1x sterile PBS. Collaged solution was incubated at 37˚C for 2 hours before being plated to 12mm coverslips. Collagen was allowed to incubate on coverslips 12 hours at 37˚C. Slips were then rinsed with 1x PBS and dried in 37˚C incubator. Slips were stored in sterile 1x PBS until use.

Vero E6 cells were plated onto collaged coated slips at 30,000 cells/slip. 24 hours after plating, cells were either left uninfected, or infected with SARS-CoV02 (WA-1) at an MOI = 0.01 and incubated for 24 hours. At that time cells were washed once with sterile 1x PBS and then fixed with 4% PFA solution (Electron Microscopy Sciences; 15710) for 10 minutes at room temperature. Following PFA fixation, cells were washed three times with 1x PBS and stored at 4˚C until use.

For immunofluorescent staining, coverslips were incubated for 2 hours at RT with a blocking buffer consisting of 1% Triton X-100, 0.5% sodium deoxycholate, 1% egg albumin and 0.05% sodium azide all suspended in 1x PBS. After 2-hour blockade, coverslips were incubated overnight at 4˚C with primary antibodies against hACE2 (goat anti-hACE2, R&D; AF933), and hAXL (rabbit anti-hAXL, Cell Signaling; C89E7). Cells were washed three times with 1x PBS for 5 minutes each. Samples with goat anti-hACE2 primary antibodies were first incubated with donkey anti-goat IgG (H+L) Alexa Fluor 568 (Invitrogen; A-11057) for 1 hour at RT. Cells were washed three times with 1x PBS for 5 minutes. Coverslips incubated with donkey anti-goat Alexa Fluor 568 were then incubated with 5% NGS (Sigma; G9023) for one hour at RT. Cells were washed three times 1x PBS then incubated for one-hour room temperature with goat anti-rabbit IgG (H+L) Alexa Fluor Plus 488 (Invitrogen; A32731TR). After washing three times for 5 minutes with 1x PBS cells were fixed to glass microscopy slides (Fisher Scientific; 12-550-15) with 12ul of Prolong Glass mounting medium (Invitrogen; P36982). Coverslips cured for 48 hours before imaging.

Image acquisition was performed on the Leica SP8 3X STED confocal microscope equipped with an HC PL APO C32 100X/1.4 oil objective lens, and LAS X software (Leica Microsystems; version 3.5.5.19976) in the Central Microscopy Research Facility at the University of Iowa. Excitation was performed using a white light laser set to 20% intensity. Depletion was performed with a 660nm laser set at 7% intensity for Alexa Fluor 488, and 775 nm with 16.5% laser for Alexa Fluor 568 fluorophore. Depletion lasers were aligned using the auto STED alignment tool in the LAS X software. Laser strength and gain were adjusted to prevent pixel saturation. Images collected with two-line averages and 2 frame accumulations. Post image analyses include deconvolution using Huygens Professional Software and the deconvolution Wizard auto functionality (Scientific Volume Imaging). Fluorescence intensity plot profiles

were created using ColorProfiler plug-in for ImageJ (Dimiter Prodanov). Pearson's correlation coefficients were performed by using freehand ROI selection tool in ImageJ to outline individual cells and performing colocalization calculations using the Colocalization Test plugin (Tony Collins et al.). Depicted are the R values for 20 cells across five separate fields imaged from one coverslip.

## Quantification and statistical analysis

Statistical analysis was completed in GraphPad Prism v9.0.2 (GraphPad Software, San Diego, CA). Quantification of flow cytometry data was completed in FlowJo v10.7.1 (Becton, Dickinson & Company, Ashland, OR). Statistical significance was defined as $p < 0.05$ and denoted by a single asterisk (*). Details regarding statistical tests used and exact values of n can be found in the corresponding figure legends. All data presented is representative of n = 3 independent experiments unless otherwise noted.

## Supporting information

**S1 Fig. PS receptors synergize with ACE2, enhancing SARS-CoV-2 infection of HEK 293T cells. A)** Representative surface staining of receptors transfected into cells. **B)** Surface expression (MFI) of proteins in mock transfected (empty vector) and transfected HEK 293T at 48 hours after transfection. Background fluorescence is shown for secondary antibodies used in experiment (α-goat or rabbit secondaries). **C)** HEK 293T cells, transfected PS receptors as noted with or without 250 ng of ACE2 were transduced with rVSV/Spike. Transduction was assessed 24 hours later via luminescence. **D)** Expression of MerTK did not affect rVSV/Spike transduction in the presence of 250 ng of transfected ACE2 plasmid. **E)** Expression of ACE2, TIM-1 or AXL did not enhance infection of VSV-luciferase/Lassa virus GP pseudovirions. HEK 293T cells were transfected with PS receptor plasmids and 50 ng of ACE2 and infected 48 hours later. Panels **C**, **D**, and **E** are shown as fold change of luciferase activity in cell lysates relative to mock transfected lysates that were set to a value of 1. Data shown are pooled from at least three independent experiments (**C**, **D** and **E**). Data represented as means ± SEM. One-Way ANOVA with multiple comparisons (**C, E**), Student's t-test (**D**); asterisks represent $p < 0.05$.
(PDF)

**S2 Fig. PS receptors interact with SARS-CoV-2 by binding to PS. A**) PS liposomes compete with PS on purified virions for binding to bavituximab. **B**) TIM-1 mutant ND115DN is highly expressed after plasmid transfection in HEK 293T cells. **C**) AXL surface expression in transfected HEK 293T cells. **D**) Purified Spike ectodomain-Fc and NTD-Fc are detected by an NTD monoclonal antibody by ELISA. **E**) All Spike-Fc proteins bind and are detected at equivalent levels of ELISA plates. Data represented as means ± SEM. Two-Way ANOVA with multiple comparisons (**A**), One-Way ANOVA with multiple comparisons (**B**); asterisks represent $p < 0.05$.
(PDF)

**S3 Fig. The route of SARS-CoV-2 entry is altered by TMPRSS2 expression.** ATPLite cytotoxicity assay in human lung cells, H1650, 24 hours following treatment with E64. Data are represented as means +/- SEM.
(PDF)

**S4 Fig. AXL has a prominent role in SARS-CoV-2 entry in Vero E6 cells. A)** ACE2, AXL and TIM-1 surface expression MFI in Vero E6 cells, as assessed by flow cytometry. Background fluorescence is shown for secondary antibodies used in experiment. **B)** Cell surface

versus intracellular ACE2 expression in Vero E6 cells. Cells were lifted, permeabilized as noted, and stained with anti-ACE2 unconjugated primary antibodies and Alexa 647 secondaries and analyzed by flow cytometry. **C)** Bemcentinib toxicity 24 hours after treatment was measured by ATPlite assay in H1650 cell line. **D)** VSV/Spike entry was measured by flow cytometry 24 hours after virus challenge of Vero E6 cells treated with bemcentinib. **E)** Vero E6 were treated with ARD5 (anti-human TIM-1 blocking antibody) 1 hour before infection with rVSV/Spike or rVSV/EBOV-GP (MOI = 0.01). Viral load was measured 24 hpi by RT-qPCR and presented normalized to the highest MOI for each rVSV. **F)** Plot profiles of ACE2 and AXL intensity are shown from STED micrographs in **Fig 4F**, representing signal intensity along the yellow lines in the merged panels. Data in **A**, **C**, **D** and **E** are shown as means ± SEM. Multiple t-tests were performed in **C** and Student's t-test was performed in **D** and **E**; asterisks represent $p < 0.05$.
(PDF)

**S5 Fig. AXL inhibition reduces SARS-CoV-2 infection in human lung cells. A)** Human lung cell lines used in these studies were stained for cell surface ACE2, AXL, TIM-1, and TMPRSS2 protein, and expression was quantified by flow cytometry. Shown are flow cytometry histograms depicting target surface staining (black line) and secondary only background (gray shade). **B)** H1650 cells were treated with 1 μM of bemcentinib and infected with one of three different variants of SARS-CoV-2: WA-1; B.1.1.7 or B.1.351 (MOI = 0.5 for all variants). RNA was isolated at 24 hpi and assessed for virus load. **C)** PS liposomes do not inhibit SARS-CoV-2 infection in Calu-3 cells. Cells were pretreated with liposomes at indicated doses, infected with SARS-CoV-2, and viral load was assessed 24 hpi. **D)** H1650 cells were infected with SARS-CoV-2 (MOI = 0.5) after treatment with the indicated concentration of camostat for 1 hour. Viral loads 24hpi were measured by RT-qPCR. **E)** Cell surface and intracellular staining of ACE2 is shown in multiple cell lines. These data are shown as frequency positive cells. Data represented as means ± SEM. Data are representative of 3 independent experiments (**B**). Student's t-test (**B**, **C**, **E**); asterisks represent $p < 0.05$.
(PDF)

**S6 Fig. AXL knockout reduces viral loads and ablates bemcentinib inhibition. A)** Knock down of AXL (AXL^low) in H1650 and HCC2302 cells were generated by lentiviral transduction of Cas9 and gRNA targeting AXL, followed by selection (no clonal isolation). These polyclonal cells are designated "AXL^low". Shown are flow cytometry histograms depicting AXL surface staining (black) and secondary only background (gray), demonstrating clear reduction of AXL expression in bulk populations of cells. **B)** H1650 AXL^low and H1650 Cas9 (parental) lines were challenged with SARS-CoV-2 at indicated MOIs for 24 hpi and viral loads assessed by RT-qPCR. **C)** H1650 parental and AXL^low lines were treated with indicated concentration of bemcentinib and challenged with SARS-CoV-2 (MOI = 0.5) and viral loads determine by RT-qPCR 24 hpi. **D-E)** As in **B-C** with HCC2302 cells. Data are pooled from at least 3 independent experiments (**B**, **C**, **D**, **E**) or are representative of at least 3 experiments (**A**). Data represented as means ± SEM. Multiple t-tests; asterisks represent $p < 0.05$.
(PDF)

**S7 Fig. A)** Bone marrow derived macrophages from C57bl6/J mice were treated as indicated and challenged with MHV (strain A59) at the indicated MOI. Viral loads were assessed 24 hpi by RT-qPCR. **B)** BMDMs were treated with indicated concentrations of bemcentinib for 1 hour, infected with MHV (MOI = 0.001) for 24 hours and viral load assessed by RT-qPCR. **C)** As in B, MHV infection of murine peritoneal macrophages (MOI = 0.001) treated with indicated concentrations of bemcentinib. **D)** Peritoneal macrophages from WT and AXL^lacz mice

were infected with MHV at multiple MOIs, and viral load was quantified 24 hpi. **E)** VSV/Spike pseudovirions were incubated with Vero E6 cells in the presence of the indicated compounds, and infection was assessed for luciferase activity 48 hours later. Data shown are 2 pooled (**D**, **E**) or representative of 3 independent (**A**, **B**, **C**) experiments. Data represented as means ± SEM. Student's t-test; asterisks represent $p < 0.05$.
(PDF)

## Acknowledgments

Kaitlyn Bohan graciously designed and created our graphical abstract image. The Cas9/BlastR lentiviral expression vector was generously provided by Aloysius Klingelhutz, PhD. The A549<sup>ACE2</sup> cell line was kindly provided by Balaji Manicassamy, PhD. Paige Richards assisted isolation of the H1650 AXL<sup>neg</sup> clone. Melinda Brindley, PhD, at the University of Georgia kindly provided rVSV/Spike-nanoluciferase.

Data presented herein were obtained at the Genomics Division of the Iowa Institute of Human Genetics which is supported, in part, by the University of Iowa Carver College of Medicine. The authors would like to acknowledge vital assistance from Chantal Allamargot, PhD. Imaging was performed using the Leica SP8 STED super resolution confocal available for use in the microscopy core.

Data presented herein were obtained at the Flow Cytometry Facility, which is a Carver College of Medicine/Holden Comprehensive Cancer Center core research facility at the University of Iowa. The facility is funded through user fees and the generous financial support of the Carver College of Medicine, Holden Comprehensive Cancer Center, and Iowa City Veteran's Administration Medical Center.

## Author Contributions

**Conceptualization:** Dana Bohan, Hanora Van Ert, Kai J. Rogers, John Minna, James B. Lorens, Wendy Maury.

**Data curation:** Dana Bohan, Hanora Van Ert, Tomasz Stokowy, Wendy Maury.

**Formal analysis:** Dana Bohan, Hanora Van Ert, Natalie Ruggio, Kai J. Rogers, Tomasz Stokowy, David Micklem, Hillel Haim, Wendy Maury.

**Funding acquisition:** Wendy Maury.

**Investigation:** Dana Bohan, Hanora Van Ert, Natalie Ruggio, Kai J. Rogers, Mohammad Badreddine, José A. Aguilar Briseño, Jonah M. Elliff, Roberth Anthony Rojas Chavez, Eleni Christakou, Petri Kursula, David Micklem, Hillel Haim, Wendy Maury.

**Methodology:** Dana Bohan, Hanora Van Ert, José A. Aguilar Briseño, Petri Kursula, David Micklem, Hillel Haim, James B. Lorens, Wendy Maury.

**Project administration:** Dana Bohan, Wendy Maury.

**Resources:** Boning Gao, Petri Kursula, David Micklem, Gro Gausdal, Hillel Haim, John Minna, James B. Lorens, Wendy Maury.

**Software:** Tomasz Stokowy.

**Supervision:** James B. Lorens, Wendy Maury.

**Validation:** Dana Bohan, Hanora Van Ert, Wendy Maury.

**Visualization:** Dana Bohan, Hanora Van Ert.

**Writing – original draft:** Dana Bohan, Wendy Maury.

**Writing – review & editing:** Dana Bohan, Hanora Van Ert, Kai J. Rogers, José A. Aguilar Briseño, Wendy Maury.

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
