## [Decision Letter · Decision Letter 0]

10 Aug 2021

Dear Dr. Maury,

Thank you very much for submitting your manuscript "Phosphatidylserine Receptors Enhance SARS-CoV-2 Infection: AXL as a Therapeutic Target for COVID-19" for consideration at PLOS Pathogens. As with all papers reviewed by the journal, your manuscript was reviewed by members of the editorial board and by several independent reviewers. In light of the reviews (below this email), we would like to invite the resubmission of a significantly-revised version that takes into account the reviewers' comments.

We cannot make any decision about publication until we have seen the revised manuscript and your response to the reviewers' comments. Your revised manuscript is also likely to be sent to reviewers for further evaluation.

Sincerely,

Sean P.J. Whelan

Associate Editor

PLOS Pathogens

Michael Diamond

Section Editor

PLOS Pathogens

Kasturi Haldar

Editor-in-Chief

PLOS Pathogens

orcid.org/0000-0001-5065-158X

Michael Malim

Editor-in-Chief

PLOS Pathogens

orcid.org/0000-0002-7699-2064

Reviewer's Responses to Questions

**Part I - Summary**

Reviewer #1: This submission provides indirect but compelling evidence that TIM and TAM receptors (or TAM adaptors) bind phosphatidyl serine (PS) on SARS-CoV-2 (SARS2) pseudoviruses and authentic viruses, and that these TIM/TAM:PS interactions facilitate virus-cell binding and ACE2-dependent virus-cell entry. The findings help to clarify the field, particularly because a prior report claims that one TAM (called AXL) binds directly to SARS spike protein domains. This submission argues against this TAM-spike binding and makes the more conventional claim of PS binding, consistent with abundant prior data on TIM/TAM interactions with PS on several diverse enveloped virus membranes. Furthermore, this submission makes the sensible claims that TIM/TAM molecules are only recognized as assisting SARS2 entry when ACE2 levels are low, and that ACE2 is still required even when TIM/TAM are abundant. The data supporting these claims are generally strong, although the levels at which TIM/TAM receptors support SARS2 entry are often fairly modest. Physiological relevance of TIM/TAM in vivo was not addressed. There are also some puzzling findings in the report (for example, TIMs support SARS2 entry in some cell types while TAM (AXL) does so in others, even though both TIMs and TAMs are all concurrently available to the virus). Hence some findings are left unexplained. However, given that identifying and characterizing SARS2 susceptibility factors are valuable in understanding the SARS2 pandemic, the report appears to be a worthy contribution to the field.

Reviewer #2: In this manuscript by Bohan and colleagues, the role of phosphatidylserine (PS) receptor in SARS-CoV-2 entry is investigated. In this study, the authors first show that, while not sufficient for infection, expression of PS receptor Axl, TIM-1, and TIM-4, but not TYRO3 or MerTK, increases SARS-CoV-2 infection of 293T cells expressing low levels of Ace2 or a wider range of level of expression of Ace2 for TIM-1. This enhancement of entry correlated with an increase in virion attachment to cells in the presence of Axl and TIM-1. The authors then show that liposomes containing PS block the enhancement of infection conferred by the expression of the PS receptor, and, using a TIM-1 mutant, that the ability of TIM-1 to bind PS is essential for its function as an entry factor. Given that the authors did not detect an interaction between Axl and SARS-CoV-2 S or the S N-terminal domain, they conclude that the enhancement of infection by the PS receptors is via interaction with exposed PS in the viral envelope. The authors then investigated whether the PS receptor-dependent enhancement of infection is restricted to one particular route of entry (surface vs endosomal). As others, they showed that co-expression of TMPRSS2 with Ace2 redirect entry from an endosomal route that is E64 sensitive to a surface entry that is E64 insensitive. The additional expression of PS receptor did not modify the entry route which was dictated by TMPRSS2 presence or absence. The presence of PS did not seem to enhance infection in the context of TMPRSS2 co-expression in 293T-Ace2, although the authors did not comment on this or explore this further. The authors then study the roles of these receptors in Vero cells, which endogenously express Ace2, Axl, and TIM-1. In these cells, PS liposomes and a specific Axl kinase inhibitor reduced infection, but a TIM-1 antibody had no effect, suggesting potential differences in PS receptor usage in different cell lines. In addition, the authors found that upon infection, colocalization of Axl and Ace2 is increased and localized in intracellular vesicles, providing some insight into a potential (additional?) mechanism of action of the PS receptors. The authors then looked at the contribution of Axl, using their kinase inhibitor, to infection of various lung cell lines. The effect of the Axl inhibitor correlated with a sensitivity to the cysteine protease inhibitor E64. Using CRISPR/Cas9, the authors generated Axl KO and found that viral growth was dramatically reduced, and residual infection was insensitive to the Axl kinase inhibitor. Finally, they show that infection by another coronavirus, MHV, is also sensitive to the Axl kinase inhibitor, suggesting a wide PS receptor usage by coronaviruses.

Overall, the study is well-conducted with the use of controls and multiple tools to dissect the role of PS receptors in SARS-CoV-2 infection. The conclusions are generally well supported although there is some overinterpretation with regards to the potential mode(s) of interaction with PS receptors. In addition, a better characterization of the roles of PS receptors in the enhancement of infection (attachment, internalization, and or trafficking to Ace2), and whether this only applies to endosomal entry would strengthen this study.

**Part II – Major Issues: Key Experiments Required for Acceptance**

Reviewer #1: 1. The AXL inhibitor bemcentinib (bem) figures prominently in the study and bem is used frequently to assess AXL function in SARS2 entry. At least in Vero cells, data appear to suggest bem is pleiotropic and it is not so clear that its antiviral activity is due entirely to AXL inhibition. This is because the bem drug is far more antiviral than PS liposomes, at least in comparing Figs 4B and 4C. On lines 225-227 it is stated that a broad-spectrum TAM inhibitor is inert, therefore bem doesn’t have off-target effects. This rationale is not understood by this reviewer. Also, the results from experiments with the bem drug do not concur. For example, Fig 4C, the bem drug effects 2.5 log10 reductions of SARS2, but in Fig 4D, the same bem drug at same conditions effects only 3-fold reduction. Also in Fig 4E, the drug has modest antiviral activity. What explains this? Finally, from Fig. 6, it is concluded that the bem antiviral effect must be due to interference with AXL, because bem is less antiviral in AXL KO cells. But here and in Fig S6C and S6E, there are still bem antiviral activities in AXL KO or AXL-low cells. Some additional statistical analyses of the bem specific antiviral effects in the different cells should be further considered. This issue of bem specificity seems to matter particularly in the MHV study, because bem is used exclusively to make the claim that MHV entry into cells is promoted by interaction with PS receptors. Might off-target antiviral effects of bem in the BMDMs and peritoneal macs explain the findings in Fig S7? Could other tests, orthogonal to the pharmacologic approach with bem, be used to buttress the claims about MHV using AXL?

2. Findings are presented in challenging ways, for example, in Fig 1, results from experiments with authentic SARS2 are intermingled with results coming from SARS2 (VSV) pseudovirus transductions. Might the data be better separated? Along these lines, in comparing all results from experiments with authentic SARS2 to those with VSV pseudo SARS2, are there significant differences in TIM/TAM support or inhibition by PS liposomes or bem? Any knowledge of different PS levels on VSV vs. authentic SARS2, or any information on Gas6 binding to either of the different virus particles?

Reviewer #2: While I also favor the hypothesis that the observed enhancement of infection by the PS receptor is via virion associated PS instead of direct interaction with S, I believe that the authors have not provided the evidence needed to conclude that it is the case (lines 328-330). For instance, the reduction in the effect of PS receptors on infection by the addition of PS liposome could be due to competition with virion associated PS, as the authors propose, or on the activation and internalization of the PS receptors which would then be unavailable to the virus regardless of how it interacts, if it interacts, with the PS receptors. The use of the TIM-1 mutant is elegant, but no surface expression is presented. The authors have used this mutant in other studies, I may have missed it, but I was unable to find surface expression data in those as well. Although the data with purified recombinant soluble proteins are convincing, given that the enhancement of infection and even the increase virus attachment to cells seem to be dependent on Ace2, would it be possible that a first interaction of S with Ace2 would be required for secondary interaction with Axl or TIM-1? I would therefore urge the authors to either provide direct evidence that PS exposed on the viral surface (on SARS-CoV-2, or even on SARS-CoV-2 S-VSV) is required for the enhancement of infection, maybe via annexin V binding and competition, or to tone down their statements.

One important aspect of the potential enhancement of infection by PS receptors that is underdeveloped in this study is whether this is a mechanism that generally applies to SARS-CoV-2 entry or only one of the potential routes of entry (endosomal in that case). This is crucial to carefully assess given that this may have implications for the in vivo relevance of this mechanism. Based on Fig. 3C and the fact that Axl did not seem to be involved in serine protease-dependent surface entry by SARS-CoV-2 in lung cells (Calu3), one could argue that this mechanism only applies to the endosomal route. Unfortunately, infection of the only other lung cell line that the authors tested that seemed to express TMPRSS2 (although its activity remains to be confirmed) was E64 sensitive, suggesting that entry was via the endosomal route. Since the mechanism of Bemcentinib remains to be determined (third point below), other ways of testing a role for PS receptors would be to study the effect of PS liposomes on infection in Calu3 cells. Alternatively, Vero-TMPRSS2 could also be tested using the multiple tools the authors have used extensively. While the importance of the endosomal entry route in SARS-CoV-2 infection and pathogenesis remains to be determined, the authors should validate in which context(s) PS receptors contribute to entry and infection.

Finally, the actual mechanism(s) of the PS receptors in entry remain to be clarified. At the beginning of the manuscript, using 293T cells, the authors argue for a role of the PS receptor in attachment and only performed binding assays. A role in enhancing surface binding would indicate that these host factors could be involved in both the surface and endosomal entry routes. Later, using Vero cells, binding assays were performed with PS liposomes and these slightly, but significantly, reduced binding. However, Bemcentinib, which is an inhibitor of Axl and binds and inhibits its intracellular kinase domain and thus is not expected to have an effect on PS binding (therefore virus attachment), blocked SARS-CoV-2 infection. Although, virus binding studies in the presence of Bemcentinib are lacking, given the known mechanism of action of the drug, this would suggest a role in a post-attachment step. The authors then show that Axl may be involved in bringing virus/Axl complexes to intracellular Ace2. However, the authors do not show that Bemcentinib prevents this colocalization. In the conclusion, the authors state that they provide “evidence that PS receptors enhance SARS-CoV-2 binding to cells and mediate internalization into endosomes”. However, without internalization and trafficking assays that specifically assess PS receptor contributions (in 293T cells expressing the PS receptors and the use of Bemcentinib for the Axl/Ace2 colocalisation studies), the authors’ conclusions are not well supported.

**Part III – Minor Issues: Editorial and Data Presentation Modifications**

Reviewer #1: 3. High ACE2 levels allow SARS2 entry to take place without much need for TMPRSS2 or PS receptors. This is expected and data are shown in Fig 3. But how are the results “related to effects of soluble ACE2 on entry” (line 175)? Some further explanation is needed here.

4. The findings of abundant intracellular ACE2 (lines 195-196 and Fig S4B) are used to make the point that PS receptors bring SARS2 into ACE2-rich endosomes (lines 229-247). The presentation of S4B and lines 195-196 is out of place and confusing. It should be moved to the relevant part of the results section.

Reviewer #2: Line 90: please replace permissive for susceptible.

Line 119 and legend of Figure 1, and other places in the manuscript: I believe that the use of “synergize” for the enhancement of infection conferred by the expression of the PS receptors is incorrect. This is a minor point, but since the PS receptors do not allow entry on their own, potentiation would be the correct terminology. I would recommend to the authors to revise the terminology used in the manuscript regarding the enhancement of infection/entry by the PS receptors.

In figure 6E, is the difference between Ace2 and Ace2+TIM-1 statistically significant?

Line 792: I am confused by the legend, I am assuming that 48 hpi, is in fact 48 hours post-transfection. In addition, the section of the method describing the binding assays with 293T cells lacks clarity (lines 623-626). I am assuming that the procedure was the same as for the Vero cells except for the incubation being at room temperature, but this should be clearly stated.

Supplemental Fig.2: the labels for panel B are not visible.

A material and method section describing how the liposomes were prepared is missing. In addition, how were the experiments performed with the liposomes is unclear. The reader is left to guess based on the figure legends which seem to indicate that a pre-treatment was performed in some cases (for instance Fig.4A), while it remains unclear in most others.

Lines 241-242: in infected cells? Should mention this clearly.

The effect of the Axl inhibitor on infection of HCC1819 (Fig.5D) is puzzling given that these cells do not seem to express Axl (Fig.S5). It is even more surprising that these cells support infection at all since Ace2 is barely expressed as well. Is it only a lack of surface expression or a lack of expression overall? The authors should discuss this discrepancy.

Also, the legend states H1819 instead of HCC1819.

PLOS authors have the option to publish the peer review history of their article (what does this mean?). If published, this will include your full peer review and any attached files.

Reviewer #1: No

Reviewer #2: No
---

## [Editor Report · Decision Letter 1]

19 Oct 2021

Dear Dr. Maury,

We are pleased to inform you that your manuscript 'Phosphatidylserine Receptors Enhance SARS-CoV-2 Infection' has been provisionally accepted for publication in PLOS Pathogens.

Best regards,

Sean P.J. Whelan

Associate Editor

PLOS Pathogens

Michael Diamond

Section Editor

PLOS Pathogens

Kasturi Haldar

Editor-in-Chief

PLOS Pathogens

orcid.org/0000-0001-5065-158X

Michael Malim

Editor-in-Chief

PLOS Pathogens

orcid.org/0000-0002-7699-2064
---

## [Editor Report · Acceptance letter]

16 Nov 2021

Dear Dr. Maury,

We are delighted to inform you that your manuscript, "Phosphatidylserine Receptors Enhance SARS-CoV-2 Infection," has been formally accepted for publication in PLOS Pathogens.

Best regards,

Kasturi Haldar

Editor-in-Chief

PLOS Pathogens

orcid.org/0000-0001-5065-158X

Michael Malim

Editor-in-Chief

PLOS Pathogens

orcid.org/0000-0002-7699-2064